# AR-ELBO: PREVENTING POSTERIOR COLLAPSE INDUCED BY OVERSMOOTHING IN GAUSSIAN VAE

## ABSTRACT

Variational autoencoders (VAEs) often suffer from posterior collapse, which is a phenomenon that the learned latent space becomes uninformative. This is related to local optima of the objective function that are often introduced by a fixed hyperparameter resembling the data variance. We suggest that this variance parameter regularizes the VAE and affects its smoothness, which is the magnitude of its gradient. An inappropriate choice of this parameter causes oversmoothness and leads to posterior collapse. This is shown theoretically by analysis on the linear approximated objective function and empirically in general cases. We propose AR-ELBO, which stands for adaptively regularized ELBO (Evidence Lower BOund). It controls the strength of regularization by adapting the variance parameter, and thus avoids oversmoothing the model. Generation models trained by proposed objectives show improved Fréchet inception distance (FID) of images generated from the MNIST and CelebA datasets.

## 1 INTRODUCTION

The variational autoencoder (VAE) framework (Kingma & Welling, 2014; Higgins et al., 2017; Zhao et al., 2019) is a popular approach to achieve generative modeling in the field of machine learning. In this framework, a model that approximates the true posterior of observation data, is learned by a joint training of encoder and decoder, which creates a stochastic mapping between the observation data and the learned deep latent space. The latent space is assumed to follow a prior distribution. The generation of a new data sample can be done by sampling the latent space and passing the sample through the decoder. It is common to assume that both the prior on the latent space and the posterior of the observation data follow a Gaussian distribution. This setup is also known as the *Gaussian VAE*. In this case, the variance of the decoder output is usually modeled as an isotropic matrix $\sigma_x^2 \mathbf{I}$ with a scalar parameter $\sigma_x^2 \geq 0$. Furthermore, in order to deal with the intractable log-likelihood of the true posterior, the evidence lower bound (ELBO) (Jordan et al., 1999) is adopted as the objective function instead.

While VAE-based generative models are usually considered to be more stable and easier to train than generative adversarial networks (Goodfellow et al., 2014), they often suffer from the problem of *posterior collapse* (Bowman et al., 2015; Sønderby et al., 2016; Alemi et al., 2017; Xu & Durrett, 2018; He et al., 2019; Razavi et al., 2019a; Ma et al., 2019), in which the latent space has little information of the input data. The phenomenon is generally mentioned as "the posterior collapses to the prior in the latent space" (Razavi et al., 2019a). Recently, several works have suggested that the variance parameter $\sigma_x^2$ in the ELBO is strongly related to posterior collapse. For example, Lucas et al. (2019) analyzed posterior collapse through the analysis on a linear VAE. It revealed that an inappropriate choice of $\sigma_x^2$ will introduce sub-optimal local optima and cause posterior collapse. Moreover, Lucas et al. (2019) reveals that contrary to the popular belief, these local optima are not introduced by replacing the log-likelihood with the ELBO, but by an excessively large $\sigma_x^2$. On the other hand, it can be shown that fixing $\sigma_x^2$ to an excessively small value leads to under-regularization of the decoder, which can cause overfitting. However, in most implementations of a Gaussian VAE, the variance parameter $\sigma_x^2$ is a fixed constant regardless of the input data and is usually 1.0. In another work, Dai & Wipf (2019) proposed a two-stage VAE and treated $\sigma_x^2$ as a training parameter. Besides the inappropriate choice of the variance parameter, posterior collapse can also induced by other causes. For example, Dai et al. (2020) found that, small nonlinear perturbation introduced in

the network architecture can also result into extra sub-optimal local minima. However, in this work we will keep our focus on the variance parameter.

We suggest that $\sigma_x^2$ affects the strength of regulation over the gradient magnitude of the decoder. We call the expected gradient magnitude the *smoothness* throughout this paper. The smaller the gradient magnitude, the smoother the model. In particular, we would like to focus on the *local smoothness* of the model, which is the smoothness evaluated within the neighborhood of the encoded latent variable of the observation data. Thus, we begin with the following hypothesis:

**Main Hypothesis.** *The value of $\sigma_x^2$ controls the regularization strength of the smoothness of the decoder. Therefore, an excessively large $\sigma_x^2$ causes oversmoothness, which results in posterior collapse.*

Following the hypothesis, the estimation of $\sigma_x^2$ should be related to properties of the approximated posterior of the latent space, such as its local smoothness. We will start with analyzing how $\sigma_x^2$ regularizes the local smoothness of the stochastic decoder and then propose new objective functions that inherently determine $\sigma_x^2$ via maximum likelihood estimation (MLE). This proposed objective function is named AR-ELBO (Adaptively Regularized ELBO), which controls the regularization strength via $\sigma_x^2$. Furthermore, several variations are derived for different parameterizations of variance parameters.

Our main contributions are listed as follows:

1. We show that our main hypothesis holds for linear approximated ELBO and empirically holds in the general case in Section 3. This also suggests that the variance parameter $\sigma_x^2$ should be estimated from properties of the approximated posterior instead of being treated as a hyperparameter.
2. We propose the AR-ELBO, an ELBO-based objective function that adaptively regularizes the smoothness of the decoder by MLE of the variance parameter $\sigma_x^2$ in Section 4. Variations of AR-ELBO for several variance parameterizations of posterior distributions are also derived. AR-ELBO prevents the model from the posterior collapse induced by oversmoothing and improves the quality of generation, which is shown in Section 5.

The organization of this paper is as follows. In Section 2, we propose a mathematical definition of **posterior collapse** in the form of mutual information. This also includes the conventional definition: "the posterior collapses to the prior in the latent space". In Section 3, the theoretical analysis and empirical support of the main hypothesis are given. We perform an analysis showing that $\sigma_x^2$ affects the smoothness of the decoder via variance parameters of the latent space learned by the encoder. In Section 4, we propose new AR-ELBO objective functions for various variance parameterizations of posterior distributions, which can relieve the decoder from being oversmoothed in the training and prevent posterior collapse. These objective functions no longer include any hyperparameters and can adaptively estimate the variance parameter $\sigma_x^2$ from the observation data. It should be noted that if we adaptively determine $\sigma_x^2$ with the proposed AR-ELBO, the strength of regularization of the decoder smoothness will gradually decrease as training progresses. In Section 5, we conduct an experiment on the MNIST and CelebA datasets, which shows that utilizing the proposed AR-ELBO with the standard Gaussian VAE can be competitive with many other variations of VAE models in most situations.

Throughout this paper, we use $a$, $\mathbf{a}$ and $\mathbf{A}$ for a scalar, a column vector and a matrix, and $\ln$ and $\log$ denote the natural logarithm and common logarithm, respectively. Our code is available from the following URL[1].

## 2 POSTERIOR COLLAPSE IN GAUSSIAN VAE

We begin with the standard formulation of the Gaussian VAE, which is the foundation of our research. A definition of posterior collapse is proposed by using mutual information (MI).

---

[1]URL hidden due to blind review.

## 2.1 GAUSSIAN VAE

Consider a data space $\mathcal{X} \subset \mathbb{R}^{d_x}$ and a sample set $\{\mathbf{x}_i\}_{i=1}^N \subset \mathcal{X}$, where $\mathbf{x}_i \sim p_{\text{data}}(\mathbf{x})$. The empirical distribution $\tilde{p}_{\text{data}}(\mathbf{x})$ on $\mathcal{X}$ can be evaluated by $\tilde{p}_{\text{data}}(\mathbf{x}) = \frac{1}{N} \sum_{n=1}^N \delta(\mathbf{x} - \mathbf{x}_n)$, where $\delta(\cdot)$ denotes the Dirac delta function. In the standard VAE framework, a latent space $\mathcal{Z} \subset \mathbb{R}^{d_z}$ is learned and the sampled latent variables $\mathbf{z} \in \mathcal{Z}$ are used to generate data samples $\mathbf{x}' \in \mathcal{X}$. Let $q_\phi(\mathbf{z}|\mathbf{x})$ and $p_\theta(\mathbf{x}|\mathbf{z})$ denote the stochastic encoder and decoder, respectively. Trainable parameters of the two neural networks are denoted as $\phi$ and $\theta$. The decoder generates data samples by $p_\theta(\mathbf{x}) := \mathbb{E}_{p(\mathbf{z})}[p_\theta(\mathbf{x}|\mathbf{z})]$, where $p(\mathbf{z})$ is the prior distribution on $\mathcal{Z}$. The encoder and decoder are jointly trained by minimizing the following objective function:

$$\mathcal{L} = -\mathbb{E}_{p_{\text{data}}(\mathbf{x})}\left[\ln p_\theta(x)\right] + \mathbb{E}_{p_{\text{data}}(\mathbf{x})}D_{\text{KL}}\left(q_\phi(\mathbf{z}|\mathbf{x}) \parallel p_\theta(\mathbf{z}|\mathbf{x})\right) + \mathbb{E}_{p_{\text{data}}(\mathbf{x})}\left[\ln p_{\text{data}}(\mathbf{x})\right]$$
$$= D_{\text{KL}}\left(p_{\text{data}}(\mathbf{x}) \parallel p_\theta(\mathbf{x})\right) + \mathbb{E}_{p_{\text{data}}(\mathbf{x})}D_{\text{KL}}\left(q_\phi(\mathbf{z}|\mathbf{x}) \parallel p_\theta(\mathbf{z}|\mathbf{x})\right). \tag{1}$$

This objective function was derived in Zhao et al. (2019), which represents everything in the form of Kullback–Leibler divergence, and is equivalent to ELBO maximization up to an additive constant.

In the context of the Gaussian VAE, the encoder and decoder are assumed to satisfy

$$q_\phi(\mathbf{z}|\mathbf{x}) = \mathcal{N}(\mathbf{z}|\boldsymbol{\mu}_\phi(\mathbf{x}), \text{diag}(\boldsymbol{\sigma}_\phi^2(\mathbf{x}))) \qquad \text{and} \qquad p_\theta(\mathbf{x}|\mathbf{z}) = \mathcal{N}(\mathbf{x}|\boldsymbol{\mu}_\theta(\mathbf{z}), \sigma_x^2\mathbf{I}). \tag{2}$$

The prior $p(\mathbf{z})$ is also assumed to be the Gaussian distribution as $p(\mathbf{z}) = \mathcal{N}(\mathbf{z}|\mathbf{0}, \mathbf{I})$. Substituting (2) into (1) while omitting terms independent of $\theta$ and $\phi$ leads to the following objective:

$$\tilde{\mathcal{J}}_{\sigma_x^2}(\theta, \phi) = \mathbb{E}_{\tilde{p}_{\text{data}}(\mathbf{x})}\left[\frac{1}{2\sigma_x^2}\mathbb{E}_{q_\phi(\mathbf{z}|\mathbf{x})}[\|\mathbf{x} - \boldsymbol{\mu}_\theta(\mathbf{z})\|_2^2] + D_{\text{KL}}(q_\phi(\mathbf{z}|\mathbf{x}) \parallel p(\mathbf{z}))\right], \tag{3}$$

which can be interpreted as the sum of the expected values of the reconstruction loss and the regularization term. In the case of a Gaussian prior and posterior, the regularization term is equal to $\frac{1}{2}\sum_{i=1}^{d_z}(\sigma_{\phi,i}^2(\mathbf{x}) + \mu_{\phi,i}(\mathbf{x})^2 - \log\sigma_{\phi,i}^2(\mathbf{x}) - 1)$.

## 2.2 POSTERIOR COLLAPSE

Posterior collapse is a major problem, where the encoder learns to map inputs to the latent space while ignoring the data distribution. In this phenomenon, the MI between input data and reconstructed data through the encoder-decoder path is reduced because the latent space has less information about the data distribution. Here, we suggest the following definition of **posterior collapse**.

**Definition 1.** *Posterior collapse is defined as the MI $\mathcal{I}(\mathbf{x}; \mathbf{x}')$ becoming nearly zero, where $\mathbf{x}' := \boldsymbol{\mu}_\theta(\mathbf{z})$ with $\mathbf{z} \sim q_\phi(\mathbf{z}|\mathbf{x})$.*

In many works (Bowman et al., 2015; Sønderby et al., 2016; Alemi et al., 2017; He et al., 2019; Razavi et al., 2019a), the phenomenon denoted as *posterior collapse* has been mathematically represented as $\mathbb{E}_{p_{\text{data}}(\mathbf{x})}D_{\text{KL}}(q_\phi(\mathbf{z}|\mathbf{x}) \parallel p(\mathbf{z})) \to 0$, which we hereafter refer to as *KL collapse* (Xu & Durrett, 2018). However, posterior collapse is not always caused by the diminished KL divergence, i.e., posterior collapse with $\mathbb{E}_{p_{\text{data}}(\mathbf{x})}D_{\text{KL}}(q_\phi(\mathbf{z}|\mathbf{x}) \parallel p(\mathbf{z})) \neq 0$ can occur. The proposed definition of posterior collapse includes *KL collapse* from the following theorem, which is proven in Appendix A.

**Theorem 2.** *$\mathcal{I}(\mathbf{x}; \mathbf{x}') \to 0$ as $\mathbb{E}_{p_{data}(\mathbf{x})}D_{\text{KL}}(q_\phi(\mathbf{z}|\mathbf{x}) \parallel p(\mathbf{z})) \to 0$ holds for any $p_\theta(\mathbf{x}|\mathbf{z})$.*

In Appendix E, it is demonstrated that posterior collapse can happen even if the KL divergence is nonzero when the posterior variance is fixed in $\mathcal{Z}$.

## 3 VARIANCE PARAMETER $\sigma_x^2$ AND THE LOCAL SMOOTHNESS

In this section, we provide mathematical and empirical support of the main hypothesis. Throughout this section, we adopt the following parameterization for the encoder for simplicity: $q_{\phi,\sigma_z^2}(\mathbf{z}|\mathbf{x}) = \mathcal{N}(\mathbf{z}|\boldsymbol{\mu}_\phi(\mathbf{x}), \sigma_z^2\mathbf{I})$, where the variance is parameterized as an isotropic matrix unlike the conventional VAE. A similar analysis on the conventional VAE can be found in Appendix B. It begins with showing that the choice of $\sigma_x^2$ affects the convergence point of $\sigma_z^2$, which is the variance parameter of the latent space. Then, we show that $\sigma_z^2$ acts as the weight of the gradient penalty, which is

implicitly included in (3). This supports the main hypothesis that the over-regulation imposed by a large $\sigma_x^2$ via $\sigma_z^2$ causes the oversmoothness of the decoder and leads to **posterior collapse**. It is also empirically supported by observing the tendencies of the convergence point of $\sigma_z^2$, the smoothness and the MI $\mathcal{I}(\mathbf{x}, \mathbf{x}')$. Ultimately, these items of evidence motivated us to develop a method that adapts $\sigma_x^2$ to prevent oversmoothing of the decoder.

### 3.1 REGULARIZATION EFFECT OF $\sigma_x^2$ IN LINEAR APPROXIMATED ELBO

The effect of $\sigma_x^2$ on the convergence point of the variance parameter $\sigma_z^2$ can be observed from two extreme cases, $\sigma_x^2 \to 0+$ and $\sigma_x^2 \to \infty$. In the first case, $\tilde{\mathcal{J}}_{\sigma_x^2}$ reduces to $\mathbb{E}_{\tilde{p}_{\text{data}}(\mathbf{x})}\mathbb{E}_{q_{\phi,\sigma_z^2}(\mathbf{z}|\mathbf{x})}[\|\mathbf{x} - \boldsymbol{\mu}_\theta(\mathbf{z})\|_2^2]$, and it becomes zero only if $\sigma_z^2 = 0$. In the second case, $\tilde{\mathcal{J}}_{\sigma_x^2}$ reduces to $\mathbb{E}_{\tilde{p}_{\text{data}}(\mathbf{x})}D_{\text{KL}}(q_{\phi,\sigma_z^2}(\mathbf{z}|\mathbf{x}) \parallel p(\mathbf{z}))$, and $\sigma_z^2$ becomes 1 at the minimum point, from $D_{\text{KL}}(q_{\phi,\sigma_z^2}(\mathbf{z}|\mathbf{x}) \parallel p(\mathbf{z})) = \frac{d_z}{2}(\sigma_z^2 - \log\sigma_z^2 - 1) + \|\boldsymbol{\mu}_\phi(\mathbf{x})\|_2^2$. This shows that a small $\sigma_x^2$ makes $\sigma_z^2$ converge to a value around 0, while a large $\sigma_x^2$ makes $\sigma_z^2$ converge to a value around 1.

If $\sigma_z^2$ is sufficiently small, as the training progresses to a certain extent, the perturbed decoding process $\boldsymbol{\mu}_\theta(\mathbf{z} + \boldsymbol{\epsilon}_z)$ around $\mathbf{z} = \boldsymbol{\mu}_\phi(\mathbf{x})$ with $\boldsymbol{\epsilon}_z \sim \mathcal{N}(\boldsymbol{\epsilon}_z|\mathbf{0}, \sigma_z^2\mathbf{I})$ can be approximated as a linear function. The ELBO can be approximated as follows by using the linear approximation of $\boldsymbol{\mu}_\theta(\cdot)$ and omitting terms independent of $\theta$ and $\phi$:

$$\tilde{\mathcal{J}}_{\sigma_x^2}(\theta, \phi, \sigma_z^2) \approx \frac{1}{2\sigma_x^2}\mathbb{E}_{\tilde{p}_{\text{data}}(\mathbf{x})}\Big[\|\mathbf{x} - \boldsymbol{\mu}_\theta(\boldsymbol{\mu}_\phi(\mathbf{x}))\|_2^2 + \sigma_z^2\|\nabla\boldsymbol{\mu}_\theta(\boldsymbol{\mu}_\phi(\mathbf{x}))\|_F^2 + 2\sigma_x^2\|\boldsymbol{\mu}_\phi(\mathbf{x})\|_2^2\Big]. \quad (4)$$

In the approximation above, $\|\cdot\|_F$ is the Frobenius norm and $\sigma_z^2$ is treated as a function parameter. The derivation of the above approximation can be found in Appendix B. Equation (4) decomposes the objective function into three terms: a reconstruction error term, gradient penalty term and $L_2$ regularization term. As one can see from (4), $\sigma_z^2$ regularizes the smoothness of the decoder by penalizing its gradient norm in training. Although the linear approximation above is derived for the simplified VAE parameterization, we also provide the linear approximation of the ELBO for the standard VAE parameterization (2) in Appendix B, where the second term in (4) becomes a weighted gradient penalty.

Summarizing the above observations shows that $\sigma_x^2$ affects the smoothness via $\sigma_z^2$, while $\sigma_z^2$ directly regularizes the smoothness. This means that if $\sigma_x^2$ is excessively large, it will cause over-regularization of the decoder and suppress $\mathcal{I}(\mathbf{z}, \mathbf{x}')(\geq \mathcal{I}(\mathbf{x}, \mathbf{x}'))$, which finally leads to posterior collapse. To avoid such over-regularization, $\sigma_x^2$ and $\sigma_z^2$ should be determined appropriately. In addition, an experiment shows that posterior collapse can be triggered by directly manipulating $\sigma_z^2$, as discussed in Appendix E.

### 3.2 EMPIRICAL STUDY ON SMOOTHNESS OF DECODER IN THE GENERAL CASE

Section 3.1 shows the impact of $\sigma_x^2$ on the regularization of the decoder smoothness through the linear approximated objective function. To support the main hypothesis in the general case, an experiment on the MNIST dataset (LeCun et al., 1998) is conducted. Several criteria are accessed to provide evidence for the regularization effect of $\sigma_x^2$ on the decoder smoothness and its consequential effect on MI $\mathcal{I}(\mathbf{x}, \mathbf{x}')$. To confirm that $\sigma_x^2$ affects the smoothness via $\sigma_z^2$, we conduct the experiment for two cases: **stochastic encoding** and **deterministic encoding**. While the stochastic encoder $q_{\phi,\sigma_z^2}(\mathbf{z}|\mathbf{x})$ is used in the former case, a VAE equipped with a deterministic encoder, i.e., $\sigma_z^2$ is fixed to zero during the training, is investigated in the latter case. Observing the difference between the two cases provides empirical support for Section 3.1. To investigate the relation between $\sigma_x^2$ and the smoothness of the decoder clearly, common generalization techniques such as batch normalization (Ioffe & Szegedy, 2015; Santurkar et al., 2018) and weight decay are not used.

**Criteria** We used several criteria to observe the impact of $\sigma_x^2$ in the experiment, such as the reconstruction error (MSE), KL divergence value $\mathbb{E}_{\tilde{p}_{\text{data}}(\mathbf{x})}D_{\text{KL}}(q_\phi(\mathbf{z}|\mathbf{x}) \parallel p(\mathbf{z}))$ and the final converged value of $\sigma_z^2$. In addition, we also estimate the local smoothness of the decoder and the MI between $\mathbf{x}$ and $\mathbf{x}'$, denoted as $\mathcal{I}(\mathbf{x}, \mathbf{x}')$. To access the local smoothness, the **expected local smoothness** (ELS) is introduced, which is the lower bound of the Lipschitz constant of the decoder. Consider a sample that is decoded with perturbation $\boldsymbol{\mu}_\theta(\boldsymbol{\mu}_\phi(\mathbf{x}) + \boldsymbol{\epsilon}_z)$, where the perturbation follows a zero-mean

Table 1: Evaluation of various criteria for different $\sigma_x^2$: the expected value of $\|\mathbf{x}' - \mathbf{x}\|_2^2$ (MSE), KL divergence, the converged value of $\sigma_z^2$, the upper bound of the MI $\mathcal{I}(\mathbf{x}'; \mathbf{z})$, the expected gap (perturbation variance $s_z^2$ are set to $10^{-2}$ and $10^{-3}$) and expected local smoothness (ELS).

| | Stochastic encoding | | | | | | | Deterministic encoding | | | | |
| $\log \sigma_x^2$ | MSE | KL | $\sigma_z^2$ | MI | Expected gap | | ELS | MSE | MI | Expected gap | | ELS |
| | | | | | $10^{-2}$ | $10^{-3}$ | | | | $10^{-2}$ | $10^{-3}$ | |
| 0.0 | **52.74** | **0.00** | **1.00** | **0.03** | 6.31e-5 | 6.35e-6 | **3.97e-4** | 5.95 | 12.5 | 74.6 | 25.3 | 7.43e+2 |
| −0.1 | 18.03 | 9.39 | 9.56e-2 | 9.7 | 1.05 | 0.108 | 6.76 | 5.69 | 14.7 | 69.1 | 22.5 | 6.82e+2 |
| −0.2 | 15.15 | 10.93 | 6.48e-2 | 12.5 | 1.30 | 0.135 | 8.34 | 5.38 | 17.9 | 63.4 | 20.7 | 6.37e+2 |
| −0.3 | 13.08 | 12.54 | 4.36e-2 | 16.0 | 1.51 | 0.157 | 9.72 | 5.37 | 21.4 | 58.1 | 17.9 | 5.78e+2 |
| −0.4 | 11.38 | 14.13 | 3.01e-2 | 20.6 | 1.77 | 0.184 | 1.14e+1 | 5.31 | 25.8 | 58.2 | 15.5 | 5.40e+2 |
| −0.5 | 10.18 | 15.30 | 2.14e-2 | 26.3 | 1.99 | 0.208 | 1.28e+1 | 5.26 | 30.6 | 53.1 | 12.9 | 4.74e+2 |
| −0.6 | 9.16 | 16.72 | 1.55e-2 | 33.2 | 2.16 | 0.227 | 1.40e+1 | 5.14 | 38.9 | 48.9 | 11.8 | 4.42e+2 |
| −0.7 | 8.25 | 18.05 | 1.11e-2 | 42.3 | 2.31 | 0.244 | 1.50e+1 | 5.17 | 46.1 | 45.5 | 10.1 | 3.98e+2 |
| −0.8 | 7.72 | 19.27 | 8.21e-3 | 52.9 | 2.40 | 0.254 | 1.56e+1 | 5.06 | 58.0 | 43.2 | 9.15 | 3.71e+2 |
| −0.9 | 7.13 | 20.55 | 5.97e-3 | 64.9 | 2.43 | 0.257 | 1.58e+1 | 4.98 | 71.9 | 39.2 | 7.83 | 3.29e+2 |
| −1.0 | 6.70 | 21.75 | 4.45e-3 | 82.3 | 2.57 | 0.272 | 1.67e+1 | 5.01 | 89.1 | 35.3 | 6.61 | 2.89e+2 |

Guassian distribution with variance $s_z^2$, $\boldsymbol{\epsilon}_z \sim \mathcal{N}(\boldsymbol{\epsilon}_z | \mathbf{0}, s_z^2 \mathbf{I})$. Let $\boldsymbol{\epsilon}_z$ and $\boldsymbol{\epsilon}_z'$ be i.i.d. random variables. We define the **expected gap** $\Delta^2(s_z^2)$ as

$$\Delta^2(s_z^2) := \mathbb{E}_{p_{\text{data}}(\mathbf{x})} \mathbb{E}_{\mathcal{N}(\boldsymbol{\epsilon}_z | \mathbf{0}, s_z^2 \mathbf{I}) \mathcal{N}(\boldsymbol{\epsilon}_z' | \mathbf{0}, s_z^2 \mathbf{I})} [\Delta^2(\mathbf{x}, \boldsymbol{\epsilon}_z, \boldsymbol{\epsilon}_z')] \tag{5}$$
$$\text{with } \Delta^2(\mathbf{x}, \boldsymbol{\epsilon}_z, \boldsymbol{\epsilon}_z') := \|\boldsymbol{\mu}_\theta(\boldsymbol{\mu}_\phi(\mathbf{x}) + \boldsymbol{\epsilon}_z) - \boldsymbol{\mu}_\theta(\boldsymbol{\mu}_\phi(\mathbf{x}) + \boldsymbol{\epsilon}_z')\|_2^2.$$

As $s_z^2$ decereases, the ratio $\Delta^2(s_z^2)/(2s_z^2)$ converges and becomes an indicator of $\mathbb{E}_{\tilde{p}_{\text{data}}}(\mathbf{x})[\|\nabla \boldsymbol{\mu}_\theta(\boldsymbol{\mu}_\phi(\mathbf{x}))\|_F^2]$, which is regularized by $\sigma_z^2$ in (4). Therefore, we can now define the ELS as

$$\mathbb{E}_{\tilde{p}_{\text{data}}(\mathbf{x})}[\|\nabla \boldsymbol{\mu}_\theta(\boldsymbol{\mu}_\phi(\mathbf{x}))\|_F^2]. \tag{6}$$

Further details of the ELS are described in Appendix C. As a reference, we estimate the upper bound of $\mathcal{I}(\mathbf{x}, \mathbf{x}')$, which is $\mathcal{I}(\mathbf{x}'; \mathbf{z})$, by Monte Carlo estimation.

**Results** Table 1 summarizes the results for different $\sigma_x^2$. In the *stochastic encoding* case, a larger $\sigma_x^2$ consistently leads to a larger $\sigma_z^2$. This results in a smaller expected gap, a smaller ELS and a lower upper bound of MI. This supports the main hypothesis that a larger $\sigma_x^2$ makes the decoder smoother. In the case of $\sigma_x^2 = 1.0$, all the criteria except MSE become nearly zero, where KL collapse and posterior collapse both occur due to the over-regularization of the smoothness of the latent space. On the other hand, in the *deterministic encoding* case, the ELS increases with $\sigma_x^2$. This is because $\sigma_x^2$ does not directly regularize the decoder via the gradient penalty as in (4). As a result, the MI upper bound does not shrink to zero even if $\sigma_x^2 = 1.0$, where posterior collapse occurs in the case of stochastic encoding. The difference in the results between the two cases clearly suggests that a large $\sigma_x^2$ triggers the oversmoothness via $\sigma_z^2$, which is consistent with the discussion in Section 3.1. These results provide empirical support of the main hypothesis as well as the discussion in Section 3.1. Further details and examples of images are shown in Appendix D.

## 3.3 DIFFICULTY OF CHOOSING AN APPROPRIATE $\sigma_x^2$

According to previous sections, a large $\sigma_x^2$ will cause oversmoothness. Therefore, we consider the case of fixing $\sigma_x^2$ to a sufficiently small value to avoid the problem. We arrive at the following theorem, whose proof can be found in Appendix F:

**Theorem 3.** *Consider the global optimum of* $\mathcal{J}_{\sigma_x^2}(\theta, \phi, \sigma_z^2)$ *w.r.t. a given* $\sigma_x^2$. *If* $\sigma_x^2 \to 0$, *then* $\sigma_z^2 \to 0$.

In Theorem 3, $\mathcal{J}_{\sigma_x^2}(\theta, \phi, \sigma_z^2)$ is optimized on the basis of the true data distribution instead of the empirical data distribution. According to the theorem, $\sigma_z^2$ converges to zero as $\sigma_x^2$ approaches zero, which leads to zero gradient penalty for the decoder as the VAE training progresses. In practice, we have no access to $p_{\text{data}}(\mathbf{x})$, but we have access to the empirical distribution $\tilde{p}_{\text{data}}(\mathbf{x})$. Theorem 3 is satisfied even when $p_{\text{data}}(\mathbf{x})$ is replaced with $\tilde{p}_{\text{data}}(\mathbf{x})$. In this case, where $\sigma_x^2$ is chosen to be small, the optimization process of $\tilde{\mathcal{J}}_{\sigma_x^2}$ will fit $p_{\theta, \sigma_x^2}(\mathbf{x})$ to the empirical distribution $\tilde{p}_{\text{data}}(\mathbf{x})$, which usually results in overfitting.

As shown above, choosing an appropriate $\sigma_x^2$ that avoids both oversmoothness and overfitting is nontrivial. Moreover, it is likely that $\sigma_x^2$ should be adapted depending on the status of training. Therefore, it is intuitive to adapt $\sigma_x^2$ instead of fixing it, which will be described in the next section.

## 4 ADAPTIVELY REGULARIZED ELBO

A modified ELBO-based objective function is proposed, which can be interpreted as an implicit update scheme that simultaneously updates $\sigma_x^2$ and the rest of the parameters. We also derive corresponding objective functions for models with different variance parameterizations.

### 4.1 ELBO WITH ADAPTIVE $\sigma_x^2$

In this subsection, we newly optimize the VAE objective function (1) w.r.t. all the parameters including $\sigma_x^2$, which is usually fixed in existing implementations. Following the process of establishing (3) but keeping the terms related to $\sigma_x^2$, we arrive at:

$$\tilde{\mathcal{J}}(\theta, \phi, \sigma_x^2) = \mathbb{E}_{\tilde{p}_{\mathrm{data}}(\mathbf{x})}\left[\frac{1}{2\sigma_x^2}\mathbb{E}_{q_\phi(\mathbf{z}|\mathbf{x})}\left[\|\mathbf{x} - \boldsymbol{\mu}_\theta(\mathbf{z})\|_2^2\right] + D_{\mathrm{KL}}\left(q_\phi(\mathbf{z}|\mathbf{x}) \parallel p(\mathbf{z})\right)\right] + \frac{d_x}{2}\ln\sigma_x^2. \quad (7)$$

From the partial derivative of $\tilde{\mathcal{J}}$ w.r.t. $\sigma_x^2$, the MLE of $\sigma_x^2$, denoted as $\hat{\sigma}_x^2$, can be evaluated with the other parameters fixed. Then, the ordinary network parameters $\theta$ and $\phi$ can be updated by optimizing (7) with the variance $\sigma_x^2$ fixed. This combination of MLE and the alternative update between $(\theta, \phi)$ and $\sigma_x^2$ guarantees that (i) if $\theta$ and $\phi$ are fixed, then there exists $\hat{\sigma}_x^2$ such that $\tilde{\mathcal{J}}(\theta, \phi, \hat{\sigma}_x^2) \leq \tilde{\mathcal{J}}(\theta, \phi, \sigma_x^2)$ and (ii) for the $\hat{\sigma}_x^2$ obtained in the previous step, there exist $\hat{\theta}$ and $\hat{\phi}$, such that $\tilde{\mathcal{J}}(\hat{\theta}, \hat{\phi}, \hat{\sigma}_x^2) \leq \tilde{\mathcal{J}}(\theta, \phi, \hat{\sigma}_x^2)$. In this respect, the convergence of the optimization is assured and the parameter $\sigma_x^2$ is kept as the MLE during the whole training stage. This inspired us to develop a weight scheduling scheme for $\hat{\sigma}_x^2$, leading to a modified ELBO-based objective function. Consider the trainable network parameters $(\theta, \phi)$ and the variance parameter $\sigma_x^2$. The update of the objective $\tilde{\mathcal{J}}(\theta, \phi, \sigma_x^2)$ is divided as

$$\sigma_x^{2(t+1)} = \frac{1}{d_x}\mathbb{E}_{\tilde{p}_{\mathrm{data}}(\mathbf{x})}\mathbb{E}_{q_{\phi^{(t)}}(\mathbf{z}|\mathbf{x})}\left[\|\mathbf{x} - \boldsymbol{\mu}_{\theta^{(t)}}(\mathbf{z})\|_2^2\right] \quad (8a)$$

$$\theta^{(t+1)}, \phi^{(t+1)} = \arg\min_{\theta, \phi}\ \tilde{\mathcal{J}}_{\sigma_x^{2(t+1)}}(\theta, \phi), \quad (8b)$$

where $t$ is the iteration index. The step updating $(\theta, \phi)$ is the same as that in the standard VAE; the step updating $\sigma_x^2$ in (8a) can be interpreted as determining an appropriate balance between the reconstruction error and the KL term in $\tilde{\mathcal{J}}_{\sigma_x^2}(\theta, \phi)$. As the learning progresses, the parameter $\sigma_x^2$ will decrease along with the MSE $\mathbb{E}_{\tilde{p}_{\mathrm{data}}(\mathbf{x})}\mathbb{E}_{q_\phi(\mathbf{z}|\mathbf{x})}[\|\mathbf{x} - \boldsymbol{\mu}_\theta(\mathbf{z})\|_2^2]$, which is consistent with the discussion in Dai & Wipf (2019).

**Proposed objective function (AR-ELBO)** The update scheme above can be further simplified by substituting (8a) into (7), which converts $\tilde{\mathcal{J}}(\theta, \phi, \hat{\sigma}_x^2)$ into

$$\tilde{\mathcal{J}}_{\mathrm{AR}}(\theta, \phi) = \frac{d_x}{2}\ln\mathbb{E}_{\tilde{p}_{\mathrm{data}}(\mathbf{x})}\mathbb{E}_{q_\phi(\mathbf{z}|\mathbf{x})}\left[\|\mathbf{x} - \boldsymbol{\mu}_\theta(\mathbf{z})\|_2^2\right] + \mathbb{E}_{\tilde{p}_{\mathrm{data}}(\mathbf{x})}D_{\mathrm{KL}}\left(q_\phi(\mathbf{z}|\mathbf{x}) \parallel p(\mathbf{z})\right), \quad (9)$$

where all constant terms w.r.t. the parameters are omitted. Optimizing (9) also makes $\sigma_x^2$ remain as the MLE during the VAE training. Moreover, (9) is equivalent to the standard Gaussian VAE plus weight balancing with (8b). This relieves the VAE from the problem of imbalance between the KL divergence term and the reconstruction loss. Also, as stated by Theorem 3, decreasing $\sigma_x^2$ also decreases $\sigma_z^2$. This gradually relieves the regularization of the ELS (6), which can be observed from (4). However, this eventually diminishes the gradient penalty; therefore, we suggest using early-stopping and learning rate scheduling to deal with this situation, which can give the decoder both appropriate smoothness and generalization capability.

### 4.2 OBJECTIVES FOR VARIOUS PARAMETERIZATIONS

In the standard VAE given by (2), the variance of the decoded distribution on $\mathcal{X}$, denoted as $\boldsymbol{\Sigma}_x$, is modeled as an identity matrix, i.e., $\boldsymbol{\Sigma}_x = \sigma_x^2\mathbf{I}$. In this case, $\sigma_x^2$ is simply a scalar value and the reconstruction objective is the same as conventional MSE and is minimized as in (9). However, out of

Table 2: Parameterizations of posterior variance in $\mathcal{X}$ and corresponding reconstruction objectives.

|  | Variance model ($\boldsymbol{\Sigma}_x$) | Reconstruction objective ($\tilde{\mathcal{J}}_{\text{rec}}(\theta, \phi, \hat{\boldsymbol{\Sigma}}_x)$) |
|---|---|---|
| (Iso-I) | $\sigma_x^2 \mathbf{I}$ | $\frac{d_x}{2} \ln \mathbb{E}_{\tilde{p}_{\text{data}}(\mathbf{x})} \mathbb{E}_{q_\phi(\mathbf{z}\|\mathbf{x})} [\|\mathbf{x} - \boldsymbol{\mu}_\theta(\mathbf{z})\|_2^2]$ |
| (Iso-D) | $\sigma_x^2(\mathbf{z}) \mathbf{I}$ | $\frac{d_x}{2} \mathbb{E}_{\tilde{p}_{\text{data}}(\mathbf{x})} \mathbb{E}_{q_\phi(\mathbf{z}\|\mathbf{x})} [\ln \|\mathbf{x} - \boldsymbol{\mu}_\theta(\mathbf{z})\|_2^2]$ |
| (Diag-I) | $\text{diag}(\boldsymbol{\sigma}_x^2)$ | $\frac{1}{2} \sum_{i=1}^{d_x} \ln \mathbb{E}_{\tilde{p}_{\text{data}}(\mathbf{x})} \mathbb{E}_{q_\phi(\mathbf{z}\|\mathbf{x})} [(x_i - \mu_{\theta,i}(\mathbf{z}))^2]$ |
| (Diag-D) | $\text{diag}(\boldsymbol{\sigma}_x^2(\mathbf{z}))$ | $\frac{1}{2} \sum_{i=1}^{d_x} \mathbb{E}_{\tilde{p}_{\text{data}}(\mathbf{x})} \mathbb{E}_{q_\phi(\mathbf{z}\|\mathbf{x})} [\ln(x_i - \mu_{\theta,i}(\mathbf{z}))^2]$ |

curiosity, we would like to explore three other variance parameterizations in addition to (2) and derive corresponding reconstruction objectives for these cases, in which the reconstruction objectives are no longer equal to MSE.

In fact, the variance $\boldsymbol{\Sigma}_x$ can not only be parameterized by an **isotropic/diagonal** matrix but also be chosen to be **independent** or **dependent** on $z$. We denote these in Table 2 as **Iso-I** (Isotropic-Independent), **Iso-D** (Isotropic-Dependent), **Diag-I** (Diagonal-Independent) and **Diag-D** (Diagonal-Dependent). The first case, Iso-I, corresponds to the standard variance model $\boldsymbol{\Sigma}_x = \sigma_x^2 \mathbf{I}$. For these parameterizations, the corresponding objectives may be summarized as

$$\tilde{\mathcal{J}}_{\text{AR}}(\theta, \phi, \hat{\boldsymbol{\Sigma}}_x) = \tilde{\mathcal{J}}_{\text{rec}}(\theta, \phi, \hat{\boldsymbol{\Sigma}}_x) + \mathbb{E}_{\tilde{p}_{\text{data}}(\mathbf{x})} D_{\text{KL}}(q(\mathbf{z}|\mathbf{x}) \| p(\mathbf{z})) \tag{10a}$$

$$\tilde{\mathcal{J}}_{\text{rec}}(\theta, \phi, \boldsymbol{\Sigma}_x) = \frac{1}{2} \mathbb{E}_{\tilde{p}_{\text{data}}(\mathbf{x})} \mathbb{E}_{q(\mathbf{z}|\mathbf{x})} \left[ \text{trace} \left( \boldsymbol{\Sigma}_x^{-1}(\mathbf{x} - \boldsymbol{\mu}_\theta(\mathbf{z}))(\mathbf{x} - \boldsymbol{\mu}_\theta(\mathbf{z}))^\top \right) + \ln |\boldsymbol{\Sigma}_x| \right]. \tag{10b}$$

By evaluating the partial derivative of $\tilde{\mathcal{J}}_{\text{rec}}$ w.r.t. $\boldsymbol{\Sigma}_x$, i.e., using its MLE, the reconstruction loss $\tilde{\mathcal{J}}_{\text{rec}}$ corresponding to each case can be derived. All the derivations can be found in Appendix G. The final reconstruction objectives with different parameterizations of $\boldsymbol{\Sigma}_x$ are listed in Table 2. It is interesting to note that the reconstruction error for each dimension in the data space has to be calculated separately for Diag-D; meanwhile, only MSE for the whole minibatch is needed in Iso-I. Considering the optimization stability in practical situations, we suggest adding a small constant, e.g., $10^{-6}$, before taking the logarithm except for in the case of Iso-I.

Although the proposed objective functions are capable of determining $\boldsymbol{\Sigma}_x$ appropriately, it should be noted that a gap still exists between the prior $p(\mathbf{z})$ and the aggregated posterior $q_\phi(\mathbf{z})$ obtained by the proposed methods. The cause can be observed from the reformulated (1) (see Appendix H):

$$\mathcal{L} = D_{\text{KL}}(p_{\text{data}}(\mathbf{x}) \| p_\theta(\mathbf{x})) + D_{\text{KL}}(p_{\text{data}}(\mathbf{x})q_\phi(\mathbf{z}|\mathbf{x}) \| q_\phi(\mathbf{z})p_\theta(\mathbf{x}|\mathbf{z})) + D_{\text{KL}}(q_\phi(\mathbf{z}) \| p(\mathbf{z})). \tag{11}$$

The first two terms in (11) can eventually become dominant in the VAE training. As a consequence, generation through sampling latent variables from the prior can cause off-distribution samples to be generated. To overcome this prior–posterior mismatch, at least two approaches can be adopted: (i) conduct another posterior estimation after the ordinary VAE training (van den Oord et al., 2017; Razavi et al., 2019b; Dai & Wipf, 2019; Ghosh et al., 2020; Morrow & Chiu, 2020) or (ii) add another regularizing term to the objective function (Makhzani et al., 2015; Tolstikhin et al., 2018; Zhao et al., 2019). The former approach is adopted in our work since it is effective (Ghosh et al., 2020) and is applicable to any VAE variation.

To summarize, the proposed AR-ELBO regularizes the Gaussian VAE with appropriate weighting for the gradient penalty without the need for an extra hyperparameter. Moreover, the remaining mismatch between the prior and posterior is mitigated by an extra pass of posterior estimation. A detailed discussion comparing the proposed objective function with previous works can be found in Appendix I.

## 5 EXPERIMENTS

We compare the proposed methods with the following models: VAE, RAE (Ghosh et al., 2020), WAE-MMD (Tolstikhin et al., 2018) and plain autoencoder (AE). The quality of generated images is evaluated using Fréchet Inception Distance (FID) (Heusel et al., 2017) on the MNIST and

Table 3: Numerical evaluation on MNIST and CelebA. The MSE of sample reconstruction is evaluated on the test set. The quality of generated samples is measured by FID for three cases: sampling the latent variables from the prior and from the estimated posterior by 2nd VAE and GMM.

| | MNIST | | | | CelebA | | | | |
|---|---|---|---|---|---|---|---|---|---|
| | MSE | FID | | | MSE | FID | | | |
| | | Prior | 2nd VAE | GMM-10 | | Prior | 2nd VAE | GMM-10 | GMM-100 |
| VAE ($\sigma_x^2 = 1.0$) | 20.25 | 55.85 | 182.64 | 58.96 | 121.91 | 55.46 | 139.32 | 54.66 | 53.94 |
| WAE-MMD | 4.34 | 22.76 | 15.00 | 13.70 | 62.66 | **52.89** | 51.32 | 43.57 | 41.88 |
| AE | 4.31 | – | 20.66 | 13.20 | 61.44 | – | 62.34 | 46.47 | 43.41 |
| RAE | **4.28** | – | 18.54 | 13.68 | 61.49 | – | 57.26 | 46.50 | 43.89 |
| RAE-GP | 4.30 | – | 18.89 | 13.71 | 61.48 | – | 54.54 | 43.63 | 41.10 |
| Iso-I w/ trainable $\mathbf{\Sigma}_x^\dagger$ | 4.28 | 20.93 | 14.83$^\ddagger$ | 13.39 | 61.42 | 63.21 | 63.12$^\ddagger$ | 51.40 | 49.61 |
| Diag-I w/ trainable $\mathbf{\Sigma}_x^\dagger$ | 5.70 | 26.08 | 18.44 | 15.72 | 62.65 | 59.66 | 54.75 | 45.86 | 43.50 |
| Iso-D w/ trainable $\mathbf{\Sigma}_x^\dagger$ | 4.45 | 27.40 | 16.38 | 13.67 | 65.29 | 195.58 | 53.13 | 50.09 | 47.31 |
| Diag-D w/ trainable $\mathbf{\Sigma}_x^\dagger$ | 5.33 | 146.43 | 26.02 | 27.90 | 85.33* | 354.20* | 217.06* | 188.30* | 188.05* |
| Ours (Iso-I w/ AR-ELBO) | 4.40 | 22.78 | 15.77 | 12.21 | 62.02 | 82.20 | 52.48 | **42.82** | 41.03 |
| Ours (Diag-I w/ AR-ELBO) | 5.35 | 24.15 | 17.18 | 13.38 | 63.51 | 87.44 | 53.60 | 45.85 | 42.83 |
| Ours (Iso-D w/ AR-ELBO) | 4.31 | 22.94 | 17.57 | 12.89 | **61.38** | 78.24 | **49.97** | 43.27 | **40.39** |
| Ours (Diag-D w/ AR-ELBO) | 6.80 | **16.64** | **10.49** | **10.05** | 70.75 | 64.40 | 55.30 | 46.63 | 45.27 |

$^\dagger$ $\mathbf{\Sigma}_x$ is learned as a trainable parameter like in Dai & Wipf (2019). However, the work does not include Diag-I, Iso-D and Diag-D parameterizations.

$^\ddagger$ 2nd VAE is the second-stage VAE proposed in Dai & Wipf (2019) after the main VAE.

* The training of Diag-D with a trainable $\mathbf{\Sigma}_x$ does not converge to a feasible local optima. Moreover, the trend of MSE diverges with that of the loss function. Therefore, the result which achieves the best MSE on the test set is reported here. On the other hand, Diag-D with the proposed AR-ELBO does not suffer from this issue.

CelebA (Liu et al., 2015) datasets with the default train/test split. Regarding the prior–posterior mismatch, three approaches are tested on all the models. The first approach is the conventional case, which samples the latent variables from the prior. The other two approaches are applied after the ordinary training. The second approach forms an aggregated posterior $q_\phi(\mathbf{z})$ by a second-stage VAE (Dai & Wipf, 2019). The third approach uses a Gaussian mixture model (GMM) with 10 to 100 components (Ghosh et al., 2020) to fit the posterior. The baseline is the standard VAE, and two methods of choosing $\sigma_x^2$ are tested: (i) $\sigma_x^2$ fixed to 1.0 as in general implementations; and (ii) $\sigma_x^2$ learned with (7) by an optimizer as an usual trainable parameter (Dai & Wipf, 2019). In RAE, its objective function is the sum of the reconstruction error, the regularization of the decoder and the $L_2$ regularization of the latent space. In RAE-GP, gradient penalty is used as the regularization of the decoder. The objective function of RAE-GP is equivalent to (4) except for that the weighting parameters are determined manually. For WAE-MMD, inverse multi-quadratic kernels with seven scales are used as proposed by Tolstikhin et al. (2018). Note that both WAE and RAE have hyperparameters in their objective functions. In contrast, the other methods, as well as the proposed objective functions, include no hyperparameters. However, regarding the major factor that affects the smoothness of the decoder, the proposed methods and VAE control smoothness by regularizing terms in their objective functions, while WAE and AE rely only on the network architecture and generalization techniques.

The latent space dimensions for MNIST and CelebA were set to $d_z = 16$ and $64$, respectively, consistent with Ghosh et al. (2020). A common network architecture, which is adopted from Chen et al. (2016) and described in Appendix J, is used for all models. In Table 3, we report the evaluation result of each method as: (i) the MSE of the reconstructed test data and (ii) the FID of generated images. The proposed methods (Diag-I, Iso-D and Diag-D) except for Iso-I do not necessarily achieve the best MSE, since the MSE is no longer the reconstruction loss for them. In the case of sampling $z \in \mathcal{Z}$ from the prior, a low FID is achieved by WAE due to the relatively strong regularization of the aggregated posterior with MMD. The images generated by the proposed method achieve the best FID score on MNIST, and is competitive on CelebA. It should be noted that the learned $\sigma_x^2$ values on MNIST and CelebA from Iso-I are $0.0056$ and $0.0050$, respectively, which are much smaller than $1.0$. Examples of reconstructed and generated images are shown in Appendix K.

As shown in Table 3, different parameterizations of variances can affect the FID greatly. In order to clearly observe the advantage of estimating $\mathbf{\Sigma}_x$ by MLE rather than estimating it as an usual trainable parameter, we examined the two approaches on all the four parameterizations (Iso-I, Iso-D, Diag-I and Diag-D): (i) solve (10b) using MLE as (9) (AR-ELBO); (ii) simply treat $\mathbf{\Sigma}_x$ as a

Table 4: Evaluation of FID scores of interpolated images on MNIST and CelebA. The interpolation ratio for each image pair is designated as (i) the mid-point; and (ii) a random-point between the two.

| | MNIST | | CelebA | |
|---|---|---|---|---|
| | Mid-point | Random-point | Mid-point | Random-point |
| VAE ($\sigma_x^2 = 1.0$) | 62.18 | 63.86 | 57.87 | 55.54 |
| WAE-MMD | 17.27 | 12.41 | **41.93** | **38.80** |
| AE | 18.49 | 12.81 | 50.35 | 45.01 |
| RAE | 17.71 | 12.99 | 48.78 | 43.97 |
| RAE-GP | 17.98 | 12.96 | 45.22 | 40.58 |
| Iso-I w/ trainable $\boldsymbol{\Sigma}_x$ | 15.66 | 12.73 | 52.01 | 48.83 |
| Diag-I w/ trainable $\boldsymbol{\Sigma}_x$ | 17.34 | 14.77 | 44.22 | 41.23 |
| Iso-D w/ trainable $\boldsymbol{\Sigma}_x$ | 17.03 | 13.33 | 48.51 | 43.97 |
| Diag-D w/ trainable $\boldsymbol{\Sigma}_x$ | 51.65 | 31.66 | 238.74 | 212.39 |
| Ours (Iso-I w/ AR-ELBO) | 14.77 | 11.27 | 42.59 | 39.19 |
| Ours (Diag-I w/ AR-ELBO) | 17.24 | 13.08 | 45.55 | 41.92 |
| Ours (Iso-D w/ AR-ELBO) | 15.30 | 12.01 | 42.49 | 38.93 |
| Ours (Diag-D w/ AR-ELBO) | **11.96** | **8.68** | 46.86 | 44.15 |

trainable parameter (Dai & Wipf, 2019). The comparative result can be obtained from the bottom eight lines of Table 3. It shows that applying AR-ELBO improves FID scores in most of the cases.

Furthermore, in order to examine the feasibility of these learned latent spaces, we also evaluated the FID scores of the images generated by interpolating two latent variables with ratios between $[0, 1]$ with all the models above. The proposed method still shows the best performance in terms of FID on MNIST and is competitive on CelebA. This result suggests the feasibility of proposed method on downstream tasks. The detail and image samples can be found in Appendix L.

## 6 CONCLUSION

We analyzed the posterior collapse phenomenon on the Gaussian VAE and investigated how strongly the variance parameter impacts the local smoothness of the decoder. The relation between the variance parameter and the local smoothness is examined both theoretically and empirically. We proposed optimization schemes to regulate the local smoothness appropriately, which leads to the prevention of posterior collapse due to oversmoothness. The proposed AR-ELBO implicitly optimizes the variance parameter to avoid over-regularizing of the smoothness. In addition, we proposed several parameterizations (Iso-D, Diag-I, Diag-D) of posterior variances, which are the extensions of the conventional VAE (Iso-I). The corresponding AR-ELBOs for these parameterizations are also derived. Our experiments show that the Gaussian VAE equipped with the proposed objective functions is competitive with other state-of-the-art models in terms of FID for both generated and interpolated images. Moreover, the proposed method remains stable in the most complicated parameterization (Diag-D). In this work, the prior–posterior mismatch was covered by extra posterior estimation methods; however, we would like to seek a thorough solution for this in the future.

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

## A  PROOF OF THEOREM 2

Let $\mathbf{x}$ be the input sample. We denote its corresponding latent space vector as $\mathbf{z}$ and the reconstructed sample as $\mathbf{x}'$. We have the following relation:

$$\mathcal{I}(\mathbf{x}; \mathbf{z}) \geq \mathcal{I}(\mathbf{x}; \mathbf{x}'), \tag{12}$$

which can be proved similarly to the proof of Lemma 5 in Appendix F. On the other hand, $\mathcal{I}(\mathbf{x}; \mathbf{z})$ can be evaluated by using the definition of the MI as

$$
\begin{aligned}
\mathcal{I}(\mathbf{x}; \mathbf{z}) &= D_{\mathrm{KL}}(\tilde{p}_{\mathrm{data}}(x)q_\phi(\mathbf{z}|\mathbf{x}) \parallel \tilde{p}_{\mathrm{data}}(x)q_\phi(\mathbf{z})) \\
&= \mathbb{E}_{\tilde{p}_{\mathrm{data}}(x)}\mathbb{E}_{q_\phi(\mathbf{z}|\mathbf{x})}[\ln q_\phi(\mathbf{z}|\mathbf{x}) - \ln q_\phi(\mathbf{z})] \\
&= \mathbb{E}_{\tilde{p}_{\mathrm{data}}(x)}D_{\mathrm{KL}}(q_\phi(\mathbf{z}|\mathbf{x}) \parallel p(\mathbf{z})) - D_{\mathrm{KL}}(q_\phi(\mathbf{z}) \parallel p(\mathbf{z})) \\
&\leq \mathbb{E}_{\tilde{p}_{\mathrm{data}}(x)}D_{\mathrm{KL}}(q_\phi(\mathbf{z}|\mathbf{x}) \parallel p(\mathbf{z})),
\end{aligned}
$$

(13)

(14)

where $\mathcal{I}(\mathbf{x}; \mathbf{z})$, $D_{\mathrm{KL}}(q_\phi(\mathbf{z}|\mathbf{x}) \parallel p(\mathbf{z}))$ and $D_{\mathrm{KL}}(q_\phi(\mathbf{z}) \parallel p(\mathbf{z}))$ are all non-negative. Inequalities (12) and (14) lead to the proof.

# B    LINEAR APPROXIMATION OF THE ELBO-BASED OBJECTIVE $\mathcal{J}_{\sigma_x^2}$

We start with parameterizing the encoder while following the assumption in (2). Given a sufficiently small perturbation with $p(\boldsymbol{\epsilon}_z) = \mathcal{N}(\mathbf{z}|\mathbf{0}, \mathrm{diag}(\boldsymbol{\sigma}_\phi^2(\mathbf{x})))$, the linear approximation of $\boldsymbol{\mu}_\theta(\cdot)$ at $\boldsymbol{\mu}_\phi(\mathbf{x})$ can be represented as

$$\boldsymbol{\mu}_\theta(\boldsymbol{\mu}_\phi(\mathbf{x}) + \boldsymbol{\epsilon}_z) = \boldsymbol{\mu}_\theta(\boldsymbol{\mu}_\phi(\mathbf{x})) + J_{\boldsymbol{\mu}_\theta}(\boldsymbol{\mu}_\phi(\mathbf{x}))\boldsymbol{\epsilon}_z, \tag{15}$$

where $J_{\boldsymbol{\mu}_\theta}(\boldsymbol{\mu}_\phi(\mathbf{x}))$ represents the Jacobian matrix of $\boldsymbol{\mu}_\theta(\mathbf{z})$ at $\mathbf{z} = \boldsymbol{\mu}_\phi(\mathbf{x})$. Substituting (15) into (3) leads to

$$\mathbb{E}_{q_\phi(\mathbf{z}|\mathbf{x})}[\|\mathbf{x} - \boldsymbol{\mu}_\theta(\mathbf{z})\|_2^2]$$
$$= \mathbb{E}_{\mathcal{N}(\mathbf{z}|\mathbf{0}, \mathrm{diag}(\boldsymbol{\sigma}_\phi^2(\mathbf{x})))}\left[\|\mathbf{x} - (\boldsymbol{\mu}_\theta(\boldsymbol{\mu}_\phi(\mathbf{x})) + J_{\boldsymbol{\mu}_\theta}(\boldsymbol{\mu}_\phi(\mathbf{x}))\boldsymbol{\epsilon}_z)\|_2^2\right]$$
$$= \|\mathbf{x} - \boldsymbol{\mu}_\theta(\boldsymbol{\mu}_\phi(\mathbf{x}))\|_2^2 + \mathbb{E}_{\mathcal{N}(\mathbf{z}|\mathbf{0}, \mathrm{diag}(\boldsymbol{\sigma}_\phi^2(\mathbf{x})))}\left[\boldsymbol{\epsilon}_z^\top J_{\boldsymbol{\mu}_\theta}(\boldsymbol{\mu}_\phi(\mathbf{x}))^\top J_{\boldsymbol{\mu}_\theta}(\boldsymbol{\mu}_\phi(\mathbf{x}))\boldsymbol{\epsilon}_z\right]$$
$$+ \underbrace{\mathbb{E}_{\mathcal{N}(\mathbf{z}|\mathbf{0}, \mathrm{diag}(\boldsymbol{\sigma}_\phi^2(\mathbf{x})))}\left[(\mathbf{x} - \boldsymbol{\mu}_\theta(\boldsymbol{\mu}_\phi(\mathbf{x})))^\top J_{\boldsymbol{\mu}_\theta}(\boldsymbol{\mu}_\phi(\mathbf{x}))\boldsymbol{\epsilon}_z\right]}_{=0}, \tag{16}$$

where the last term is zero under the assumption that the perturbation is sufficiently small. The expectation in the second right-hand-side term can be evaluated as

$$\mathbb{E}_{\mathcal{N}(\mathbf{z}|\mathbf{0}, \mathrm{diag}(\boldsymbol{\sigma}_\phi^2(\mathbf{x})))}\left[\boldsymbol{\epsilon}_z^\top J_{\boldsymbol{\mu}_\theta}(\boldsymbol{\mu}_\phi(\mathbf{x}))^\top J_{\boldsymbol{\mu}_\theta}(\boldsymbol{\mu}_\phi(\mathbf{x}))\boldsymbol{\epsilon}_z\right]$$
$$= \mathrm{trace}\left(\mathbb{E}_{\mathcal{N}(\mathbf{z}|\mathbf{0}, \mathrm{diag}(\boldsymbol{\sigma}_\phi^2(\mathbf{x})))}\left[\boldsymbol{\epsilon}_z\boldsymbol{\epsilon}_z^\top\right] J_{\boldsymbol{\mu}_\theta}(\boldsymbol{\mu}_\phi(\mathbf{x}))^\top J_{\boldsymbol{\mu}_\theta}(\boldsymbol{\mu}_\phi(\mathbf{x}))\right)$$
$$= \mathrm{trace}\left(\mathrm{diag}(\boldsymbol{\sigma}_\phi^2(\mathbf{x})) J_{\boldsymbol{\mu}_\theta}(\boldsymbol{\mu}_\phi(\mathbf{x}))^\top J_{\boldsymbol{\mu}_\theta}(\boldsymbol{\mu}_\phi(\mathbf{x}))\right)$$
$$= \sum_{i=1}^{d_x}\sum_{j=1}^{d_z}\sigma_{\phi,j}^2(\mathbf{x})\left(\left.\frac{\partial\mu_{\theta,i}(\mathbf{z})}{\partial z_j}\right|_{z=\boldsymbol{\mu}_\phi(\mathbf{z})}\right)^2, \tag{17}$$

which can be interpreted as the gradient penalty for the decoder weighted by $\boldsymbol{\sigma}_\phi^2(\mathbf{x})$. By substituting the above result into (3), its linear approximation can be obtained as

$$\tilde{\mathcal{J}}_{\sigma_x^2}(\theta, \phi) \approx \frac{1}{2\sigma_x^2}\mathbb{E}_{\tilde{p}_{\mathrm{data}}(\mathbf{x})}\left[\|\mathbf{x} - \boldsymbol{\mu}_\theta(\boldsymbol{\mu}_\phi(\mathbf{x}))\|_2^2\right.$$
$$\left. + \sum_{i=1}^{d_x}\sum_{j=1}^{d_z}\sigma_{\phi,j}^2(\mathbf{x})\left(\left.\frac{\partial\mu_{\theta,i}(\mathbf{z})}{\partial z_j}\right|_{\mathbf{z}=\boldsymbol{\mu}_\phi(\mathbf{x})}\right)^2 + 2\sigma_x^2\|\boldsymbol{\mu}_\phi(\mathbf{x})\|_2^2\right]. \tag{18}$$

In the case of the simplified parameterization described in Section 3, the second right-hand-side term in (18) can be further reduced to $\sigma_z^2\|\nabla\boldsymbol{\mu}_\theta(\boldsymbol{\mu}_\phi(\mathbf{x}))\|_F^2$. In the simplified case, the perturbation follows a multivariate i.i.d. Gaussian distribution, $\boldsymbol{\epsilon}_z \sim \mathcal{N}(\mathbf{z}|\mathbf{0}, \sigma_z^2\mathbf{I})$. Under this assumption, we have

$$\mathbb{E}_{p(\boldsymbol{\epsilon}_z)}\left[\boldsymbol{\epsilon}_z^\top J_{\boldsymbol{\mu}_\theta}(\boldsymbol{\mu}_\phi(\mathbf{x}))^\top J_{\boldsymbol{\mu}_\theta}(\boldsymbol{\mu}_\phi(\mathbf{x}))\boldsymbol{\epsilon}_z\right] = \mathbb{E}_{p(\boldsymbol{\epsilon}_z)}\left[\boldsymbol{\epsilon}_z^\top\left(\sum_{i=1}^{d_z}\lambda_i\mathbf{u}_i(\mathbf{x})\mathbf{u}_i(\mathbf{x})^\top\right)\boldsymbol{\epsilon}_z\right]$$
$$= \sum_{i=1}^{d_z}\lambda_i\mathbf{u}_i(\mathbf{x})^\top\mathbb{E}_{p(\boldsymbol{\epsilon}_z)}[\boldsymbol{\epsilon}_z\boldsymbol{\epsilon}_z^\top]\mathbf{u}_i(\mathbf{x})$$
$$= 2\sigma_z^2\sum_{i=1}^{d_z}\lambda_i, \tag{19}$$

where $\lambda_i$ is the $i$th eigenvalue of $J_{\boldsymbol{\mu}_\theta}(\boldsymbol{\mu}_\phi(\mathbf{x}))J_{\boldsymbol{\mu}_\theta}(\boldsymbol{\mu}_\phi(\mathbf{x}))^\top$, which is a symmetrical positive definite matrix, and the corresponding eigenvectors are $(\mathbf{u}_i(\mathbf{x}))_{i=1}^{d_z}$. Following the simplified assumption, the second right-hand-side term in (16) now becomes $\mathbb{E}_{\mathcal{N}(\mathbf{z}|\mathbf{0},\sigma_z^2\mathbf{I})}\left[\boldsymbol{\epsilon}_z^\top J_{\boldsymbol{\mu}_\theta}(\boldsymbol{\mu}_\phi(\mathbf{x}))^\top J_{\boldsymbol{\mu}_\theta}(\boldsymbol{\mu}_\phi(\mathbf{x}))\boldsymbol{\epsilon}_z\right]$. Combining (19) and the fact that $\sum_{i=1}^{d_z}\lambda_i = \mathrm{trace}(J_{\boldsymbol{\mu}_\theta}(\boldsymbol{\mu}_\phi(\mathbf{x}))J_{\boldsymbol{\mu}_\theta}(\boldsymbol{\mu}_\phi(\mathbf{x}))^\top) =$

$\|\nabla\boldsymbol{\mu}_\theta(\boldsymbol{\mu}_\phi(\mathbf{x}))\|_2^2$, we can finally obtain the following linear approximation for the simplified parameterization:

$$
\begin{aligned}
\mathbb{E}_{\mathcal{N}(\mathbf{z}|\mathbf{0},\sigma_z^2\mathbf{I})} &\left[\boldsymbol{\epsilon}_z^\top J_{\boldsymbol{\mu}_\theta}(\boldsymbol{\mu}_\phi(\mathbf{x}))^\top J_{\boldsymbol{\mu}_\theta}(\boldsymbol{\mu}_\phi(\mathbf{x}))\boldsymbol{\epsilon}_z\right] \\
&= \mathbb{E}_{\mathcal{N}(\mathbf{z}|\mathbf{0},\sigma_z^2\mathbf{I})}\left[\operatorname{trace}\left(J_{\boldsymbol{\mu}_\theta}(\boldsymbol{\mu}_\phi(\mathbf{x}))^\top J_{\boldsymbol{\mu}_\theta}(\boldsymbol{\mu}_\phi(\mathbf{x}))\right)\right] \\
&= \sigma_z^2 \sum_{i=1}^{d_x}\sum_{j=1}^{d_z}\left(\left.\frac{\partial\mu_{\theta,i}(\mathbf{z})}{\partial z_j}\right|_{\mathbf{z}=\boldsymbol{\mu}_\phi(\mathbf{z})}\right)^2 \\
&= \sigma_z^2\|\nabla\boldsymbol{\mu}_\theta(\boldsymbol{\mu}_\phi(\mathbf{x}))\|_F^2.
\end{aligned}
\tag{20}
$$

## C  EXPECTED LOCAL SMOOTHNESS OF DECODER

Here, we describe the relation between the expected local smoothness $\mathbb{E}_{\tilde{p}_{\text{data}}(\mathbf{x})}[\|\nabla\boldsymbol{\mu}_\theta(\boldsymbol{\mu}_\phi(\mathbf{x}))\|_F^2]$ and the expected gap $\Delta^2(s_z^2)$. First, consider the relation

$$
\Delta^2(\mathbf{x},\boldsymbol{\epsilon}_z,\boldsymbol{\epsilon}_z') := \|\boldsymbol{\mu}_\theta(\boldsymbol{\mu}_\phi(\mathbf{x})+\boldsymbol{\epsilon}_z) - \boldsymbol{\mu}_\theta(\boldsymbol{\mu}_\phi(\mathbf{x})+\boldsymbol{\epsilon}_z')\|_2^2 = K_\theta(\boldsymbol{\mu}_\phi(\mathbf{x}),\boldsymbol{\epsilon}_z,\boldsymbol{\epsilon}_z')^2\|\boldsymbol{\epsilon}_z-\boldsymbol{\epsilon}_z'\|_2^2,
\tag{21}
$$

with the perturbation $\boldsymbol{\epsilon}_z$ following the Gaussian distribution $\mathcal{N}(\boldsymbol{\epsilon}_z|\mathbf{0},s_z^2\mathbf{I})$. Applying the expectation operator to (21) leads to

$$
\Delta^2(\mathbf{x},\boldsymbol{\epsilon}_z,\boldsymbol{\epsilon}_z') = \mathbb{E}_{p(\boldsymbol{\epsilon}_z,\boldsymbol{\epsilon}_z')}\left[K_\theta(\boldsymbol{\mu}_\phi(\mathbf{x}),\boldsymbol{\epsilon}_z,\boldsymbol{\epsilon}_z')^2\|\boldsymbol{\epsilon}_z-\boldsymbol{\epsilon}_z'\|_2^2\right]
\tag{22}
$$

$$
\leq \mathbb{E}_{p(\boldsymbol{\epsilon}_z,\boldsymbol{\epsilon}_z')}\left[K_\theta(\boldsymbol{\mu}_\phi(\mathbf{x}),\boldsymbol{\epsilon}_z,\boldsymbol{\epsilon}_z')^2\right]\mathbb{E}_{p(\boldsymbol{\epsilon}_z,\boldsymbol{\epsilon}_z')}\left[\|\boldsymbol{\epsilon}_z-\boldsymbol{\epsilon}_z'\|_2^2\right]
\tag{23}
$$

$$
=: 2K_\theta^2(\boldsymbol{\mu}_\phi(\mathbf{x}),s_z^2)d_z s_z^2,
\tag{24}
$$

where $p(\boldsymbol{\epsilon}_z,\boldsymbol{\epsilon}_z') := \mathcal{N}(\boldsymbol{\epsilon}_z|\mathbf{0},s_z^2\mathbf{I})\mathcal{N}(\boldsymbol{\epsilon}_z'|\mathbf{0},s_z^2\mathbf{I})$, $\boldsymbol{\epsilon}_z-\boldsymbol{\epsilon}_z'\sim\mathcal{N}(\boldsymbol{\epsilon}_z-\boldsymbol{\epsilon}_z'|\mathbf{0},2s_z^2\mathbf{I})$ and $K_\theta^2(\boldsymbol{\mu}_\phi(\mathbf{x})) := \mathbb{E}_{p(\boldsymbol{\epsilon}_z,\boldsymbol{\epsilon}_z')}\left[K_\theta(\boldsymbol{\mu}_\phi(\mathbf{x}),\boldsymbol{\epsilon}_z,\boldsymbol{\epsilon}_z')^2\right]$. Note that in (23), we assume that $K_\theta^2(\boldsymbol{\mu}_\phi(\mathbf{x}))$ is independent of $\boldsymbol{\epsilon}_z$ and $\boldsymbol{\epsilon}_z'$. Consider the case that the variance $s_z^2$ is sufficiently small to approximate $\boldsymbol{\mu}_\theta(\mathbf{z})$ linearly around $z = \boldsymbol{\mu}_\phi(\mathbf{x})$, which is perturbed with variance $s_z^2$. In such a case, $K_\theta(\boldsymbol{\mu}_\phi(\mathbf{x}),\boldsymbol{\epsilon}_z,\boldsymbol{\epsilon}_z')$ is independent of $\boldsymbol{\epsilon}_z$ and $\boldsymbol{\epsilon}_z'$, which fits the assumption in (23). Under this local linearity assumption, $K_\theta^2(\boldsymbol{\mu}_\phi(\mathbf{x}))$ is bounded as

$$
K_\theta^2(\boldsymbol{\mu}_\phi(\mathbf{x}),s_z^2) \leq K_\theta^2,
\tag{25}
$$

where $K_\theta$ denotes the Lipschitz constant of the decoder.

Following the assumption, $K_\theta^2(\boldsymbol{\mu}_\phi(\mathbf{x}),s_z^2)$ can be formulated by invoking (15) as

$$
\begin{aligned}
K_\theta^2(\boldsymbol{\mu}_\phi(\mathbf{x}),s_z^2) &= \frac{\mathbb{E}_{p(\boldsymbol{\epsilon}_z,\boldsymbol{\epsilon}_z')}[\|\boldsymbol{\mu}_\theta(\boldsymbol{\mu}_\phi(\mathbf{x})+\boldsymbol{\epsilon}_z)-\boldsymbol{\mu}_\theta(\boldsymbol{\mu}_\phi(\mathbf{x})+\boldsymbol{\epsilon}_z')\|_2^2]}{\mathbb{E}_{p(\boldsymbol{\epsilon}_z,\boldsymbol{\epsilon}_z')}[\|\boldsymbol{\epsilon}_z-\boldsymbol{\epsilon}_z'\|_2^2]} \\
&= \frac{\mathbb{E}_{p(\boldsymbol{\epsilon}_z,\boldsymbol{\epsilon}_z')}[(\boldsymbol{\epsilon}_z-\boldsymbol{\epsilon}_z')^\top J_{\boldsymbol{\mu}_\theta}(\boldsymbol{\mu}_\phi(\mathbf{x}))^\top J_{\boldsymbol{\mu}_\theta}(\boldsymbol{\mu}_\phi(\mathbf{x}))(\boldsymbol{\epsilon}_z-\boldsymbol{\epsilon}_z')]}{2d_z s_z^2} \\
&= \frac{\operatorname{trace}\left(\mathbb{E}_{p(\boldsymbol{\epsilon}_z,\boldsymbol{\epsilon}_z')}[(\boldsymbol{\epsilon}_z-\boldsymbol{\epsilon}_z')(\boldsymbol{\epsilon}_z-\boldsymbol{\epsilon}_z')^\top]J_{\boldsymbol{\mu}_\theta}(\boldsymbol{\mu}_\phi(\mathbf{x}))^\top J_{\boldsymbol{\mu}_\theta}(\boldsymbol{\mu}_\phi(\mathbf{x}))\right)}{2d_z s_z^2} \\
&= \frac{\operatorname{trace}\left(J_{\boldsymbol{\mu}_\theta}(\boldsymbol{\mu}_\phi(\mathbf{x}))^\top J_{\boldsymbol{\mu}_\theta}(\boldsymbol{\mu}_\phi(\mathbf{x}))\right)}{d_z}.
\end{aligned}
\tag{26}
$$

Applying the expectation operator to (26) leads to

$$
\begin{aligned}
K_\theta^2(s_z^2) &:= \mathbb{E}_{\tilde{p}_{\text{data}}}\left[K_\theta^2(\boldsymbol{\mu}_\phi(\mathbf{x}),s_z^2)\right] \\
&= \frac{\mathbb{E}_{\tilde{p}_{\text{data}}(\mathbf{x})}\left[\operatorname{trace}\left(J_{\boldsymbol{\mu}_\theta}(\boldsymbol{\mu}_\phi(\mathbf{x}))^\top J_{\boldsymbol{\mu}_\theta}(\boldsymbol{\mu}_\phi(\mathbf{x}))\right)\right]}{d_z} \\
&= \frac{\mathbb{E}_{\tilde{p}_{\text{data}}}\left[\|\nabla\boldsymbol{\mu}_\theta(\boldsymbol{\mu}_\phi(\mathbf{x}))\|_F^2\right]}{d_z}.
\end{aligned}
\tag{27}
$$

Finally, combining (24) and (27) yields the following connection between the expected gap and the expected local smoothness:

$$
\Delta^2(s_z^2) = 2\mathbb{E}_{\tilde{p}_{\text{data}}}\left[\|\nabla\boldsymbol{\mu}_\theta(\boldsymbol{\mu}_\phi(\mathbf{x}))\|_F^2\right]s_z^2.
\tag{28}
$$

# D  EXPERIMENTAL DETAILS FOR SECTION 3.2

## D.1  EXPERIMENTAL SETUP

In the experiment, the model is trained with the Adam optimizer with a learning rate of $10^{-3}$. The dimension of the latent space is set to 8. We run 200 epochs with a minibatch size of 64 for all $\sigma_x^2$. We use the following DNN architectures for the encoder and decoder, respectively:

$$
\begin{aligned}
x \in \mathbb{R}^{28 \times 28} &\to \mathrm{Conv}_{64} \to \mathrm{ReLU} && \text{size of } (64, 14, 14) \\
&\to \mathrm{Conv}_{128} \to \mathrm{ReLU} \to \mathrm{Reshape}(128, 7, 7) && \text{size of } (128, 7, 7) \\
&\to \mathrm{Flatten} \to \mathrm{FC}_{1024} \to \mathrm{ReLU} \\
&\to \mathrm{FC}_{16}, \\
z \in \mathbb{R}^{16} &\to \mathrm{FC}_{1024} \to \mathrm{ReLU} \\
&\to \mathrm{FC}_{128 \times 7 \times 7} \to \mathrm{ReLU} && \text{size of } (128, 7, 7) \\
&\to \mathrm{ConvT}_{64} \to \mathrm{ReLU} && \text{size of } (64, 14, 14) \\
&\to \mathrm{ConvT}_{1} \to \mathrm{Sigmoid} && \text{size of } (1, 28, 28).
\end{aligned}
$$

Here, $\mathrm{FC}_k$, $\mathrm{Conv}_k$, $\mathrm{ConvT}_k$ and ReLU denote the fully connected layer mapping to $\mathbb{R}^k$, the convolutional layer mapping to $k$ channels, the transpose convolutional layer mapping to $k$ channels and the rectified linear units (ReLU), respectively. The 3-tuple $(\mathrm{channels}, \mathrm{height}, \mathrm{width})$ in the right column represents the output shape of each layer. In all the $\mathrm{Conv}_k$ and $\mathrm{ConvT}_k$ layers, $4 \times 4$ convolutional filters are used with a common stride of $(2, 2)$.

Regarding the evaluation of criteria, MSE and KL are evaluated on the training set because the aim of the experiment is to validate the relation between $\sigma_z^2$ and the smoothness of the decoder. The upper bound of the MI is obtained by calculating

$$
-\mathbb{E}_{\tilde{p}_{\mathrm{data}}(\mathbf{x})} \mathbb{E}_{q_{\phi, \sigma_z^2}(\mathbf{z}|\mathbf{x})} \left[ \ln \mathbb{E}_{\tilde{p}_{\mathrm{data}}(\mathbf{x}')} \exp \left( -\frac{1}{2\sigma_z^2} \|\mathbf{z} - \boldsymbol{\mu}_\phi(\mathbf{x}')\|_2^2 \right) \right] - \frac{d_x}{2} \tag{29}
$$

for each minibatch and then taking their mean, where the batch size is $10{,}000$ for all the evaluations.

## D.2  SAMPLES OF GENERATED IMAGES AND T-SNE VISUALIZATION OF LATENT SPACES

Figure 1 shows several images decoded from $\boldsymbol{\mu}_\phi(\mathbf{x}) + \boldsymbol{\epsilon}_z$ with $\boldsymbol{\epsilon}_z \sim \mathcal{N}(\boldsymbol{\epsilon}_z | \mathbf{0}, s_z^2 \mathbf{I})$ for the cases with $\sigma_x^2 = 1.0$ and 0.1. Posterior collapse can be observed from these blurry images decoded from the *stochastic encoding* case with $\sigma_x^2 = 1.0$. This is due to the removal of batch normalization, which makes $\sigma_x^2 = 1.0$ become an inappropriate choice. However, if $\sigma_x^2$ is determined or adapted appropriately such as by using the proposed method, posterior collapse will not happen. In the other settings, the tendency of how the image changes with the perturbation is similar, as shown in Table 1.

| | Stochastic encoding | | Deterministic encoding | |
|---|---|---|---|---|
| | $s_z^2 = 10^{-2}$ | $s_z^2 = 10^{-3}$ | $s_z^2 = 10^{-2}$ | $s_z^2 = 10^{-3}$ |
| $\sigma_x^2 = 1.0$ |  |  |  |  |
| $\sigma_x^2 = 0.1$ |  |  |  |  |

Figure 1: Images in red boxes are the original images sampled from the MNIST dataset. The images in blue boxes are reconstructed by $\boldsymbol{\mu}_\theta(\boldsymbol{\mu}_\phi(\mathbf{x}))$. The other images are decoded from neighbor points of $\boldsymbol{\mu}_\phi(\mathbf{x})$, which are perturbed by $\boldsymbol{\epsilon}_z \sim \mathcal{N}(\boldsymbol{\epsilon}_z | \mathbf{0}, s_z^2 \mathbf{I})$.

The latent spaces are also visualized via t-SNE (Maaten & Hinton, 2008) in Figure 2. The dots with different colors represent the latent vectors encoded from images of different labels (numbers), and the pink dots are the sampling points generated from the prior $p(\mathbf{z})$. As mentioned earlier, to observe the effect of $\sigma_x^2$ clearly, we remove batch normalization, which usually helps prevent posterior collapse to a certain extent. As a result, the latent space with $\sigma_x^2 = 1.0$ completely collapses and $q_\phi(z)$ approaches $p(z)$ as shown in Figures 1 and 2(a). In this case, both KL collapse and posterior collapse occur.

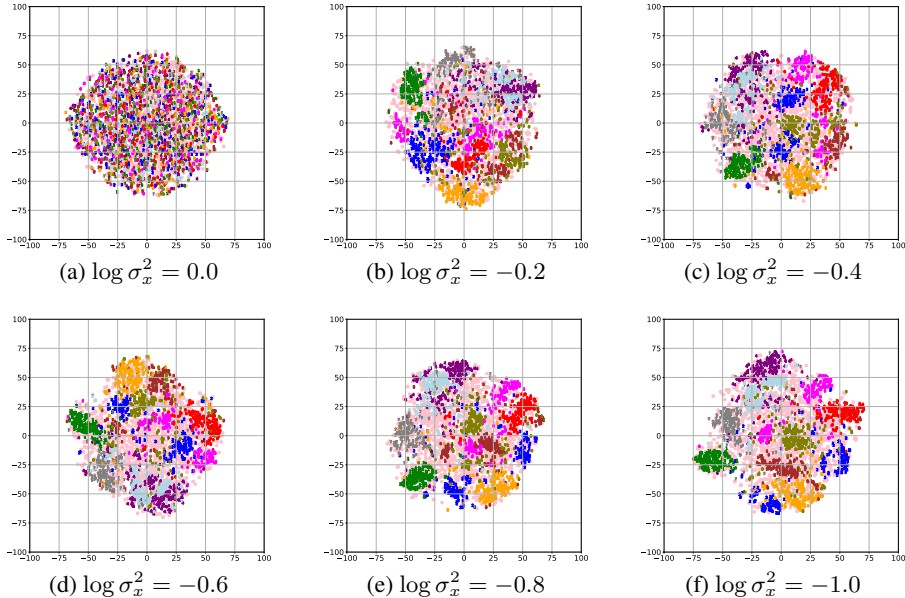

(a) $\log \sigma_x^2 = 0.0$     (b) $\log \sigma_x^2 = -0.2$     (c) $\log \sigma_x^2 = -0.4$

(d) $\log \sigma_x^2 = -0.6$     (e) $\log \sigma_x^2 = -0.8$     (f) $\log \sigma_x^2 = -1.0$

Figure 2: Visualization of latent space via t-SNE. Pink dots are sampling points generated from the prior $p(\mathbf{z})$.

Table 5: Evaluation of various criteria for different $\sigma_z^2$. These criteria are the expected value of $\|\mathbf{x}' - \mathbf{x}\|_2^2$ (MSE), KL divergence, the upper bound of MI $\mathcal{I}(\mathbf{x}', \mathbf{z})$, the expected gap (the perturbation variance $s_z^2$ is set to $10^{-2}$ and $10^{-3}$) and expected local smoothness (ELS).

| $\log \sigma_z^2$ | MSE | KL | MI | Expected gap $10^{-2}$ | Expected gap $10^{-3}$ | ELS |
|---|---|---|---|---|---|---|
| 1.0 | **52.74** | **26.79** | **8.0e-3** | 6.20e-6 | 6.15e-7 | **3.81e-5** |
| 0.9 | **52.74** | **19.48** | **6.8e-3** | 7.19e-6 | 7.11e-7 | **4.42e-5** |
| 0.8 | 22.96 | 134.83 | 8.5e+1 | 2.03e-2 | 2.03e-3 | 1.27e-1 |
| 0.7 | 20.50 | 139.09 | 1.0e+2 | 2.37e-2 | 2.37e-3 | 1.48e-1 |
| 0.6 | 19.30 | 132.61 | 1.2e+2 | 2.87e-2 | 2.87e-3 | 1.80e-1 |
| 0.5 | 17.38 | 132.05 | 1.7e+2 | 3.44e-2 | 3.44e-3 | 2.16e-1 |
| 0.4 | 15.91 | 128.96 | 1.9e+2 | 3.94e-2 | 3.94e-3 | 2.47e-1 |
| 0.3 | 14.63 | 125.97 | 2.3e+2 | 4.50e-2 | 4.50e-3 | 2.82e-1 |
| 0.2 | 13.50 | 122.07 | 2.8e+2 | 5.16e-2 | 5.16e-3 | 3.23e-1 |
| 0.1 | 12.40 | 118.89 | 3.1e+2 | 5.83e-2 | 5.81e-3 | 3.67e-1 |
| 0.0 | 11.71 | 112.50 | 3.6e+2 | 6.79e-2 | 6.80e-3 | 4.26e-1 |
| −0.1 | 10.97 | 107.32 | 4.0e+2 | 7.53e-2 | 7.55e-3 | 4.74e-1 |
| −0.2 | 10.23 | 103.87 | 4.2e+2 | 8.69e-2 | 8.70e-3 | 5.46e-1 |
| −0.3 | 9.63 | 98.48 | 4.6e+2 | 9.84e-2 | 9.88e-3 | 6.18e-1 |
| −0.4 | 9.12 | 93.86 | 5.2e+2 | 1.12e-1 | 1.12e-2 | 7.05e-1 |
| −0.5 | 8.71 | 88.35 | 5.2e+2 | 1.25e-1 | 1.26e-2 | 7.88e-1 |
| −0.6 | 8.26 | 83.68 | 5.9e+2 | 1.42e-1 | 1.43e-2 | 8.94e-1 |
| −0.7 | 7.82 | 79.70 | 6.6e+2 | 1.62e-1 | 1.62e-2 | 1.02 |
| −0.8 | 7.55 | 74.75 | 7.1e+2 | 1.80e-1 | 1.80e-2 | 1.13 |
| −0.9 | 7.26 | 70.63 | 7.3e+2 | 2.04e-1 | 2.05e-2 | 1.29 |
| −1.0 | 7.05 | 66.14 | 7.6e+2 | 2.28e-1 | 2.30e-2 | 1.45 |

## E  FIXING THE POSTERIOR VARIANCE OF LATENT SPACE

From the previous sections, we know that $\sigma_z^2$ affects the smoothness via $\sigma_x^2$. However, it would be interesting to see what will happen if $\sigma_z^2$ is fixed while $\sigma_x^2$ is optimized. In this experiment, the variance parameter $\sigma_z^2$ is fixed while $\sigma_x^2$ is optimized with the AR-ELBO (9) under the parameterization in Section 3. The other settings remain the same as those in Section 3.2. We evaluate the numerical results for different $\sigma_z^2$ with the criteria listed in Section 3.2. According to Table 5, the tendencies of the expected gap and ELS show that a large $\sigma_z^2$ makes the decoder smoother, which is consistent with the discussion in Section 3.1. However, the tendency of the KL divergence is different from that in Section 3.2. Although a larger $\sigma_z^2$ consistently leads to a smaller MI, and eventually the MI collapses to zero; the KL divergence still remains far from zero, which means that posterior collapse

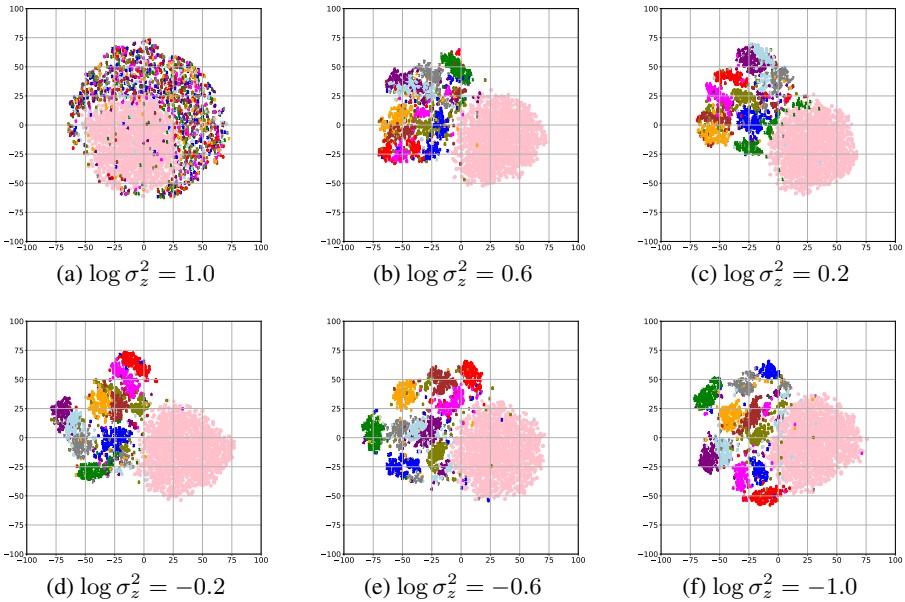

Figure 3: Visualization of latent space via t-SNE. Pink dots are sampling points generated from the prior $p(\mathbf{z})$.

can happen without KL collapse. This phenomenon can be visually confirmed by observing the t-SNE plot in Figure 3. The cause of this phenomenon can be roughly reasoned from the linear approximated ELBO (4), in which $\sigma_z^2$ directly affects the gradient penalty and causes oversmoothness. It should be pointed out that the strength of $L_2$ regularization in (4) is gradually decreased with decreasing $\sigma_x^2$; therefore, it does not dominate the whole objective function. As a result, the mean of the approximated posterior $q_\phi(z)$ is far from the mean of the prior $p(z)$ (which is $\mathbf{0}$), and therefore $D_{\mathrm{KL}}(q_\phi(z) \| p(z))$ in (13) does not diminish to zero.

## F PROOF OF THEOREM 3

According to Theorem 4 in Dai & Wipf (2019), we know that

$$\lim_{\sigma_x^2 \to 0} \mathbb{E}_{p_{\mathrm{data}}(\mathbf{x})} \mathbb{E}_{q_{\phi, \sigma_z^2}(\mathbf{z}|\mathbf{x})} \left[ \|\mathbf{x} - \boldsymbol{\mu}_\theta(\mathbf{z})\|_2^2 \right] = 0, \tag{30}$$

which also leads to $\hat{\sigma}_x^2 \to 0$ ($\sigma_x^2 \to 0$). Here, $\hat{\sigma}_x^2$ is estimated through MLE and is given by

$$\hat{\sigma}_x^2 = \frac{1}{d_x} \mathbb{E}_{\tilde{p}_{\mathrm{data}}(\mathbf{x})} \mathbb{E}_{q_{\phi, \sigma_z^2}(\mathbf{z}|\mathbf{x})} \left[ \|\mathbf{x} - \boldsymbol{\mu}_\theta(\mathbf{z})\|_2^2 \right]. \tag{31}$$

To prove Theorem 3, we need the following auxiliary theorem:

**Theorem 4.** *In the training stage of VAE, we have $\sigma_z^2 \to 0$ ($\hat{\sigma}_x^2 \to 0$).*

First, we state the two lemmas with proofs.

**Lemma 5.** *In a VAE, $\mathcal{I}(\mathbf{x}, \mathbf{x}') \leq \mathcal{I}(\mathbf{z}, \mathbf{z}_e)$ always holds, where $\mathbf{z}_e$ is the encoded latent variable $\mathbf{z}_e = \boldsymbol{\mu}_\phi(\mathbf{x})$ with $x \sim p_{data}(\mathbf{x})$.*

*Proof.* The data processing flow of the VAE is $\mathbf{x} \to \mathbf{z}_e \to \mathbf{z} \to \mathbf{x}'$; $\mathbf{z}_e = \boldsymbol{\mu}_\phi(\mathbf{x})$, $\mathbf{z} = \mathbf{z}_e + \boldsymbol{\epsilon}_z$, and $\mathbf{x}' = \boldsymbol{\mu}_\theta(\mathbf{z})$, where $\boldsymbol{\epsilon}_z \sim \mathcal{N}(\boldsymbol{\epsilon}_z | \mathbf{0}, \sigma_z^2 \mathbf{I})$. The MI $\mathcal{I}(\mathbf{x}; \mathbf{z}, \mathbf{x}')$ can be represented as

$$\mathcal{I}(\mathbf{x}; \mathbf{z}, \mathbf{x}') = \mathcal{I}(\mathbf{x}; \mathbf{x}') + \mathcal{I}(\mathbf{x}; \mathbf{z}|\mathbf{x}') \tag{32}$$

$$= \mathcal{I}(\mathbf{x}; \mathbf{z}) + \mathcal{I}(\mathbf{x}; \mathbf{x}'|\mathbf{z}). \tag{33}$$

Since $\mathbf{x}$ and $\mathbf{x}'$ are conditionally independent on the given $z$, it follows that $\mathcal{I}(\mathbf{x}; \mathbf{x}'|\mathbf{z}) = 0$. From the non-negativity of MI, we have $\mathcal{I}(\mathbf{x}; \mathbf{z}) \geq \mathcal{I}(\mathbf{x}; \mathbf{x}')$. Repeating the same procedure for $\mathcal{I}(\mathbf{x}; \mathbf{z}_e, \mathbf{z})$ leads to the proof. $\square$

**Lemma 6.** *The MI between $\mathbf{x}$ and $\mathbf{x}'$ diverges to positive infinity as $\sigma_x^2 \to 0$, where $\mathbf{x}'$ is obtained from $\mathbf{x} \sim p_{data}(\mathbf{x})$ as $\mathbf{x}' = \boldsymbol{\mu}_\theta(\boldsymbol{\mu}_\phi(\mathbf{x}) + \boldsymbol{\epsilon}_z)$.*

*Proof.* A lower bound of $\mathcal{I}(\mathbf{x}; \mathbf{x}')$ is

$$
\begin{aligned}
\mathcal{I}(\mathbf{x}; \mathbf{x}') &= D_{\mathrm{KL}}\left(p_{\mathrm{data}}(\mathbf{x})p_{\theta,\phi}(\mathbf{x}'|\mathbf{x}) \parallel p_{\mathrm{data}}(\mathbf{x})p_{\theta,\phi}(\mathbf{x}')\right) \\
&= \mathbb{E}_{p_{\mathrm{data}}(\mathbf{x})}\mathbb{E}_{p_{\theta,\phi}(\mathbf{x}'|\mathbf{x})}\left[\ln p_{\theta,\phi}(\mathbf{x}'|\mathbf{x}) - \ln p_{\theta,\phi}(\mathbf{x}')\right] \\
&= \mathcal{H}\left[p_{\theta,\phi}(\mathbf{x}')\right] - \mathbb{E}_{p_{\mathrm{data}}(\mathbf{x})}\mathcal{H}\left[p_{\theta,\phi}(\mathbf{x}'|\mathbf{x})\right] \\
&\geq \mathcal{H}\left[p_{\theta,\phi}(\mathbf{x}')\right] - \mathbb{E}_{p_{\mathrm{data}}(\mathbf{x})}H(\hat{\sigma}_x^2\mathbf{I}),
\end{aligned}
\tag{34}
$$

where $p_{\theta,\phi}(\mathbf{x}'|\mathbf{x}) := \mathbb{E}_{q_\phi(\mathbf{z}|\mathbf{x})}[p_\theta(\mathbf{x}'|\mathbf{z})]$ and $p_{\theta,\phi}(\mathbf{x}') := \mathbb{E}_{p_{\mathrm{data}}(\mathbf{x})}\mathbb{E}_{q_\phi(\mathbf{z}|\mathbf{x})}[p_\theta(\mathbf{x}'|\mathbf{z})]$. Here, we denote the differential entropy of the Gaussian with variance $\hat{\sigma}_x^2\mathbf{I}$ as

$$
H(\hat{\sigma}_x^2\mathbf{I}) := \frac{1}{2}\ln(2\pi e\hat{\sigma}_x^{2d_x}).
\tag{35}
$$

Since $\hat{\sigma}_x^2 \to 0$ as $\sigma_x^2 \to 0$, $\mathcal{H}[p_{\theta,\phi}(\mathbf{x}')] \to \mathcal{H}[p_{\mathrm{data}}(\mathbf{x})]$ and $H(\hat{\sigma}_x^2) \to -\infty$ in the inequality of (34). Therefore, $\mathcal{I}(\mathbf{x}; \mathbf{x}') \to \infty$ as $\sigma_x^2 \to 0$. $\qquad\square$

Now we prove Theorem 3. The MI $\mathcal{I}(\mathbf{z}; \mathbf{z}_\mathrm{e})$ satisfies

$$
\begin{aligned}
\mathcal{I}(\mathbf{z}; \mathbf{z}_\mathrm{e}) &= D_{\mathrm{KL}}\left(q_\phi(\mathbf{z}_\mathrm{e})q_{\sigma_z^2}(\mathbf{z}|\mathbf{z}_\mathrm{e}) \parallel q_\phi(\mathbf{z}_\mathrm{e})q_{\phi,\sigma_z^2}(\mathbf{z})\right) \\
&= \mathbb{E}_{q_\phi(\mathbf{z}_\mathrm{e})}\mathbb{E}_{q_{\sigma_z^2}(\mathbf{z}|\mathbf{z}_\mathrm{e})}\left[\ln q_{\sigma_z^2}(\mathbf{z}|\mathbf{z}_\mathrm{e}) - \ln q_{\phi,\sigma_z^2}(\mathbf{z})\right] \\
&= \mathcal{H}\left[q_{\phi,\sigma_z^2}(\mathbf{z})\right] - \mathbb{E}_{q_\phi(\mathbf{z}_\mathrm{e})}\mathcal{H}\left[q_{\sigma_z^2}(\mathbf{z}|\mathbf{z}_\mathrm{e})\right] \\
&\leq H(\boldsymbol{\Sigma}_{\phi,\sigma_z^2}) - H(\sigma_z^2\mathbf{I}) \\
&= \frac{d_z}{2}\ln\left(\frac{\det(\boldsymbol{\Sigma}_{\phi,\sigma_z^2})}{\sigma_z^2}\right),
\end{aligned}
\tag{36}
$$

where $\boldsymbol{\Sigma}_{\phi,\sigma_z^2}$ denotes the variance of $q_{\phi,\sigma_z^2}(\mathbf{z})$. Invoking Lemma 5 and (36) leads to

$$
\mathcal{I}(\mathbf{x}; \mathbf{x}') \leq \frac{d_z}{2}\ln\left(\frac{\det(\boldsymbol{\Sigma}_{\phi,\sigma_z^2})}{\sigma_z^2}\right).
\tag{37}
$$

Now, consider $\sigma_z^2 \nrightarrow 0$ as $\mathcal{L} \to 0$. According to Lemma 6, it follows that $\det(\boldsymbol{\Sigma}_{\phi,\sigma_z^2}) \to +\infty$, which contradicts the fact that $q_{\phi,\sigma_z^2}(\mathbf{z}) \to p(\mathbf{z})$. Thus, we must have $\sigma_z^2 \to 0$ as $\sigma_x^2$ converges to zero.

## G  DERIVATION OF PROPOSED OBJECTIVES

Here, we derive the objectives listed in Table 2. Consider an arbitrary $\boldsymbol{\Sigma}_x$ without any condition. The MLE of $\boldsymbol{\Sigma}_x$, $\hat{\boldsymbol{\Sigma}}_x$, can be obtained by

$$
\hat{\boldsymbol{\Sigma}}_x = \mathbb{E}_{\tilde{p}_{\mathrm{data}}(\mathbf{x})}\mathbb{E}_{q_\phi(\mathbf{z}|\mathbf{x})}\left[(\mathbf{x} - \boldsymbol{\mu}_\theta(\mathbf{z}))(\mathbf{x} - \boldsymbol{\mu}_\theta(\mathbf{z}))^\top\right].
\tag{38}
$$

From the partial derivative of $\tilde{\mathcal{J}}_{\mathrm{rec}}(\theta, \phi, \boldsymbol{\Sigma}_x)$ w.r.t. $\boldsymbol{\Sigma}_x$, we have

$$
\frac{\partial\tilde{\mathcal{J}}_{\mathrm{rec}}(\theta, \phi, \boldsymbol{\Sigma}_x)}{\partial\boldsymbol{\Sigma}_x} = \frac{1}{2}\left(\mathbb{E}_{\tilde{p}_{\mathrm{data}}}\mathbb{E}_{q_\phi(\mathbf{z}|\mathbf{x})}\left[(\mathbf{x} - \boldsymbol{\mu}_\theta(\mathbf{z}))(\mathbf{x} - \boldsymbol{\mu}_\theta(\mathbf{z}))^\top\right] + \boldsymbol{\Sigma}_x^{-1}\right).
\tag{39}
$$

The MLE of $\hat{\boldsymbol{\Sigma}}_x$ and the objectives for the different parameterizations are described in the following.

### G.1  ISO-I

For Iso-I, the MLE of $\sigma_x^2$ can be given as

$$
\hat{\sigma}_x^2 = \frac{1}{d_x}\mathbb{E}_{\tilde{p}_{\mathrm{data}}(\mathbf{x})}\mathbb{E}_{q_\phi(\mathbf{z}|\mathbf{x})}\left[\|\mathbf{x} - \boldsymbol{\mu}_\theta(\mathbf{z})\|_2^2\right].
\tag{40}
$$

Substituting (40) into (7) leads to

$$
\begin{aligned}
\tilde{\mathcal{J}}_{\mathrm{AR}}(\theta, \phi, \hat{\sigma}_x^2) &= \mathbb{E}_{\tilde{p}_{\mathrm{data}}(\mathbf{x})}\left[\frac{1}{2\hat{\sigma}_x^2}\mathbb{E}_{q_\phi(\mathbf{z}|\mathbf{x})}[\|\mathbf{x} - \boldsymbol{\mu}_\theta(\mathbf{z})\|_2^2] + D_{\mathrm{KL}}(q_\phi(\mathbf{z}|\mathbf{x}) \| p(\mathbf{z}))\right] + \frac{d_x}{2}\ln\hat{\sigma}_x^2 \\
&= \frac{d_x}{2} + \mathbb{E}_{\tilde{p}_{\mathrm{data}}(\mathbf{x})}D_{\mathrm{KL}}(q_\phi(\mathbf{z}|\mathbf{x}) \| p(\mathbf{z})) \\
&\quad + \frac{d_x}{2}\ln\mathbb{E}_{\tilde{p}_{\mathrm{data}}(\mathbf{x})}\mathbb{E}_{q_\phi(\mathbf{z}|\mathbf{x})}\left[\|\mathbf{x} - \boldsymbol{\mu}_\theta(\mathbf{z})\|_2^2\right] - \frac{d_x}{2}\ln d_x,
\end{aligned}
\tag{41}
$$

where the first and fourth terms are constants and thus omitted in (9).

## G.2   Iso-D

First, substitute $\boldsymbol{\Sigma}_x = \sigma_x^2(\mathbf{z})\mathbf{I}$ into (10b):

$$
\tilde{\mathcal{J}}_{\mathrm{rec}}(\theta, \phi, \boldsymbol{\Sigma}_x) = \mathbb{E}_{\tilde{p}_{\mathrm{data}}(\mathbf{x})}\mathbb{E}_{q_\phi(\mathbf{z}|\mathbf{x})}\left[\frac{1}{2\sigma_x^2(\mathbf{z})}\|\mathbf{x} - \boldsymbol{\mu}_\theta(\mathbf{z})\|_2^2 + \frac{d_x}{2}\ln\sigma_x^2(\mathbf{z})\right].
\tag{42}
$$

Also, we know that the MLE of $\sigma_x^2(\mathbf{z})$ is

$$
\hat{\sigma}_x^2(\mathbf{z}) = \frac{1}{d_x}\|\mathbf{x} - \boldsymbol{\mu}_\theta(\mathbf{z})\|_2^2.
\tag{43}
$$

Substituting (43) into (42) leads to the reconstruction objective of Iso-D:

$$
\begin{aligned}
\tilde{\mathcal{J}}_{\mathrm{rec}}(\theta, \phi, \hat{\boldsymbol{\Sigma}}_x) &= \mathbb{E}_{\tilde{p}_{\mathrm{data}}(\mathbf{x})}\mathbb{E}_{q_\phi(\mathbf{z}|\mathbf{x})}\left[\frac{1}{2\hat{\sigma}_x^2(\mathbf{z})}\|\mathbf{x} - \boldsymbol{\mu}_\theta(\mathbf{z})\|_2^2 + \frac{d_x}{2}\ln\hat{\sigma}_x^2(\mathbf{z})\right] \\
&= \frac{d_x}{2} + \frac{d_x}{2}\mathbb{E}_{\tilde{p}_{\mathrm{data}}(\mathbf{x})}\mathbb{E}_{q_\phi(\mathbf{z}|\mathbf{x})}\left[\ln\|\mathbf{x} - \boldsymbol{\mu}_\theta(\mathbf{z})\|_2^2\right] - \frac{d_x}{2}\ln d_x.
\end{aligned}
\tag{44}\tag{45}
$$

## G.3   Diag-I

First, substitute $\boldsymbol{\Sigma}_x = \mathrm{diag}(\boldsymbol{\sigma}_x^2)$ into (10b):

$$
\tilde{\mathcal{J}}_{\mathrm{rec}}(\theta, \phi, \boldsymbol{\Sigma}_x) = \mathbb{E}_{\tilde{p}_{\mathrm{data}}(\mathbf{x})}\left[\sum_{i=1}^{d_x}\frac{1}{2\sigma_{x,i}^2}\mathbb{E}_{q_\phi(\mathbf{z}|\mathbf{x})}\left[(x_i - \mu_{\theta,i}(\mathbf{z}))^2\right]\right] + \sum_{i=1}^{d_x}\frac{1}{2}\ln\sigma_{x,i}^2.
\tag{46}
$$

Also, we know that the MLE of $\sigma_{x,i}^2$ is

$$
\hat{\sigma}_{x,i}^2 = \mathbb{E}_{\tilde{p}_{\mathrm{data}}(\mathbf{x})}\mathbb{E}_{q_\phi(\mathbf{z}|\mathbf{x})}\left[(x_i - \mu_{\theta,i}(\mathbf{z}))^2\right].
\tag{47}
$$

Substituting (47) into (46) leads to the reconstruction objective for Diag-I:

$$
\begin{aligned}
\tilde{\mathcal{J}}_{\mathrm{rec}}(\theta, \phi, \hat{\boldsymbol{\Sigma}}_x) &= \mathbb{E}_{\tilde{p}_{\mathrm{data}}(\mathbf{x})}\left[\sum_{i=1}^{d_x}\frac{1}{2\hat{\sigma}_{x,i}^2}\mathbb{E}_{q_\phi(\mathbf{z}|\mathbf{x})}\left[(x_i - \mu_{\theta,i}(\mathbf{z}))^2\right]\right] + \sum_{i=1}^{d_x}\frac{1}{2}\ln\hat{\sigma}_{x,i}^2 \\
&= \frac{d_x}{2} + \frac{1}{2}\sum_{i=1}^{d_x}\ln\mathbb{E}_{\tilde{p}_{\mathrm{data}}(\mathbf{x})}\mathbb{E}_{q_\phi(\mathbf{z}|\mathbf{x})}\left[(x_i - \mu_{\theta,i}(\mathbf{z}))^2\right].
\end{aligned}
\tag{48}\tag{49}
$$

## G.4   Diag-D

First, substitute $\boldsymbol{\Sigma}_x = \mathrm{diag}(\boldsymbol{\sigma}_x^2(\mathbf{z}))$ into (10b):

$$
\tilde{\mathcal{J}}_{\mathrm{rec}}(\theta, \phi, \boldsymbol{\Sigma}_x) = \mathbb{E}_{\tilde{p}_{\mathrm{data}}(\mathbf{x})}\mathbb{E}_{q_\phi(\mathbf{z}|\mathbf{x})}\left[\sum_{i=1}^{d_x}\left(\frac{1}{2\sigma_{x,i}^2(\mathbf{z})}(x_i - \mu_{\theta,i}(\mathbf{z}))^2 + \frac{1}{2}\ln\sigma_{x,i}^2(\mathbf{z})\right)\right].
\tag{50}
$$

Also, we know that the MLE of $\sigma_{x,i}^2(\mathbf{z})$ is

$$
\hat{\sigma}_{x,i}^2(\mathbf{z}) = (x_i - \mu_{\theta,i}(\mathbf{z}))^2.
\tag{51}
$$

Substituting (51) into (50) leads to the reconstruction objective for Diag-D:

$$\tilde{\mathcal{J}}_{\text{rec}}(\theta, \phi, \hat{\boldsymbol{\Sigma}}_x) = \mathbb{E}_{\tilde{p}_{\text{data}}(\mathbf{x})}\mathbb{E}_{q_\phi(\mathbf{z}|\mathbf{x})}\left[\sum_{i=1}^{d_x}\left(\frac{1}{2\hat{\sigma}_{x,i}^2(\mathbf{z})}(x_i - \mu_{\theta,i}(\mathbf{z}))^2 + \frac{1}{2}\ln\hat{\sigma}_{x,i}^2(\mathbf{z})\right)\right] \qquad (52)$$

$$= \frac{d_x}{2} + \frac{1}{2}\sum_{i=1}^{d_x}\mathbb{E}_{\tilde{p}_{\text{data}}(\mathbf{x})}\mathbb{E}_{q_\phi(\mathbf{z}|\mathbf{x})}\left[\ln\left(x_i - \mu_{\theta,i}(\mathbf{z})\right)^2\right]. \qquad (53)$$

## H   Derivation of (11)

The KL divergence terms of (1) can be represented as

$$\mathbb{E}_{p_{\text{data}}(\mathbf{x})}D_{\text{KL}}(q_\phi(\mathbf{z}|\mathbf{x}) \parallel p_\theta(\mathbf{z}|\mathbf{x})) = \mathbb{E}_{p_{\text{data}}(\mathbf{x})q_\phi(\mathbf{z}|\mathbf{x})}[\ln q_\phi(\mathbf{z}|\mathbf{x}) - \ln p_\theta(\mathbf{z}|\mathbf{x})] \qquad (54)$$

$$= \mathbb{E}_{p_{\text{data}}(\mathbf{x})q_\phi(\mathbf{z}|\mathbf{x})}\left[\ln\frac{p_{\text{data}}(\mathbf{x})q_\phi(\mathbf{z}|\mathbf{x})}{p(\mathbf{z})p_\theta(\mathbf{x}|\mathbf{z})}\right]$$

and

$$\mathbb{E}_{p_{\text{data}}(\mathbf{x})q_\phi(\mathbf{z}|\mathbf{x})}\left[\ln q_\phi(\mathbf{z}) - \ln p(\mathbf{z})\right] = D_{\text{KL}}(q_\phi(\mathbf{z}) \parallel p(\mathbf{z})). \qquad (55)$$

Substituting the two equations above into (1), then $\mathcal{L}$ can be reformulated into (11).

## I   Related works

To the best of our knowledge, Lucas et al. (2019) were among the first to suggest that posterior collapse may be caused by a sub-optimal $\sigma_x^2$. In the past, one of the common approaches for dealing with posterior collapse was to anneal the weight of the KL term in the ELBO. The first such attempt was KL annealing (Bowman et al., 2015). Bowman et al. (2015) introduced a weighting coefficient on the KL term in the cost function during training. The weighting scheduling is determined in advance, e.g., the weight increases monotonically (Bowman et al., 2015; Sønderby et al., 2016) or changes cyclically (Fu et al., 2019) as the training progresses.

The weighting coefficient also appears in Higgins et al. (2017), and is interpreted as a hyperparameter that controls the information capacity of the latent space. The suggested value for such hyperparameter is larger than 1. This is analogous to setting $\sigma_x^2$ larger than the MLE value $\hat{\sigma}_x^2$ for (7) and (8a), which enforces a stronger smoothness in exchange of better latent space disentanglement. The differences between Higgins et al. (2017) and the proposed method are: (i) $\sigma_x^2$ changes between every minibatch; and (ii) the estimation of $\sigma_x^2$ is aimed to prevent the oversmoothness.

Shao et al. (2020) proposed ControlVAE, which combined control theory with the VAE, and applied PI/PID control to determine the weight on the KL term. Although applying control theory to the weighting of the KL term makes it possible to reflect the status of the optimization, ControlVAE needs extra hyperparameters to be tuned in advance. On the other hand, our method can be interpreted as automatic KL annealing that estimates $\sigma_x^2$ through MLE without the need of tuning an extra hyperparameter.

Ghosh et al. (2020) interpreted the stochastic autoencoder with the reparameterization trick as noise injection process and proposed replacing such a mechanism with an explicit regularized autoencoder (RAE). RAE regularizes its decoder in several ways: $L_2$ regularization, a gradient penalty (Gulrajani et al., 2017) and spectral normalization (Miyato et al., 2018). As discussed in Section 3.1, if $\sigma_z^2$ is sufficiently small, the ELBO can also be approximately represented as a sum of three losses (4), which correspond to the terms included in the basic RAE objective function. The approximated objective function (4) can be obtained when RAE with a gradient penalty (RAE-GP) is used and tuned appropriately.

Dai & Wipf (2019) optimized $\sigma_x^2$ using an optimizer, which is the most similar approach to our Iso-I model in that (7) is used as an objective function. Our proposed method provides simplified objective functions that enable the variance parameter to be optimized automatically and guarantee that $\sigma_x^2$ decreases as the reconstruction loss decreases, enabling the gradient penalty to be gradually weakened.

Comparing our work with these works showed that the proposed AR-ELBO and its variations are also capable of regularizing the Gaussian VAE via weighting of the gradient penalty. In the standard parameterization such as (2), the second term in (4) is a weighted gradient penalty, which makes it possible to regularize each dimension of the latent space differently according to the property of the input data. In addition, the combination of the proposed objective functions together with standard Gaussian VAE allows implicit gradient regularization on the decoder with lower computational cost than that of explicitly adding the gradient penalty to the objective function, such as in RAE.

## J  DETAILS OF EXPERIMENTAL SETUP IN SECTION 5

In this experiment, the Adam optimizer (Kingma & Ba, 2015) is used and the maximum number of epochs is set to 100 for MNIST and 70 for CelebA. The learning rates are 0.001 for MNIST and 0.0002 for CelebA. A minibatch size of 64 is used. All the FID[2] values are evaluated with $10,000$ generated samples.

For the posterior estimation by the second-stage VAE, we adopt the same networks for the encoder and decoder as those in Dai & Wipf (2019). For GMM fitting, we use the same settings as those in Ghosh et al. (2020). Experimental details including the network architectures for each dataset are described in the following.

### J.1  MNIST

We construct the encoder and decoder for the MNIST dataset using the architecture in Chen et al. (2016). The encoder is constructed as

$$x \in \mathbb{R}^{28 \times 28} \to \text{Conv}_{64} \to \text{ReLU} \qquad\qquad \text{size of } (64, 14, 14)$$
$$\to \text{Conv}_{128} \to \text{ReLU} \to \text{Reshape}(128, 7, 7) \qquad \text{size of } (128, 7, 7)$$
$$\to \text{Flatten} \to \text{FC}_{1024} \to \text{BatchNorm} \to \text{ReLU}$$
$$\to \text{FC}_{16 \times 2}.$$

The decoder is constructed as

$$z \in \mathbb{R}^{16} \to \text{FC}_{1024} \to \text{BatchNorm} \to \text{ReLU}$$
$$\to \text{FC}_{128 \times 7 \times 7} \to \text{BatchNorm} \to \text{ReLU} \qquad \text{size of } (128, 7, 7)$$
$$\to \text{ConvT}_{64} \to \text{BatchNorm} \to \text{ReLU} \qquad \text{size of } (64, 14, 14)$$
$$\to \text{ConvT}_1 \to \text{Sigmoid} \qquad\qquad \text{size of } (1, 28, 28).$$

In all the $\text{Conv}_k$ layers and all the $\text{ConvT}_k$ layers except for the last, $5 \times 5$ convolutional filters with stride $(2, 2)$ are used. The difference between this architecture and those used in Appendix D.1 is whether batch normalization is applied or not. Although in the original work of Chen et al. (2016), the discriminator used leaky ReLU (lReLU), we adopt ReLU for the encoder part, which improves the performance for all the models evenly.

### J.2  CELEBA

The CelebA images are preprocessed with center cropping of $140 \times 140$, then resized to $64 \times 64$ as described in Tolstikhin et al. (2018) and Ghosh et al. (2020). It should be noted that the size of cropping differs among the previous works, and it markedly affects the FID score. We choose the above cropping size as is the largest among the related works and seems to be the most difficult case for image generation. Moreover, Tolstikhin et al. (2018) and Ghosh et al. (2020) also used this cropping size. Similarly to in the previous section, the encoder and decoder are constructed on the basis of the discriminator and generator for CelebA used in Chen et al. (2016). The encoder is

---

[2]We used the PyTorch version of the FID implementation from `https://github.com/mseitzer/pytorch-fid` for all the models. However, the result may slightly differ from that obtained with the Tensor-Flow implementation `https://github.com/bioinf-jku/TTUR`.

constructed as

$$x \in \mathbb{R}^{64 \times 64} \to \mathrm{Conv}_{128} \to \mathrm{ReLU} \qquad\qquad \text{size of } (128, 32, 32)$$
$$\to \mathrm{Conv}_{256} \to \mathrm{BatchNorm} \to \mathrm{ReLU} \qquad \text{size of } (256, 16, 16)$$
$$\to \mathrm{Conv}_{512} \to \mathrm{BatchNorm} \to \mathrm{ReLU} \qquad \text{size of } (512, 8, 8)$$
$$\to \mathrm{Conv}_{1024} \to \mathrm{BatchNorm} \to \mathrm{ReLU} \qquad \text{size of } (1024, 4, 4)$$
$$\to \mathrm{Flatten} \to \mathrm{FC}_{64 \times 2}.$$

The decoder is constructed as

$$z \in \mathbb{R}^{64} \to \mathrm{FC}_{8 \times 8 \times 1024}$$
$$\to \mathrm{ConvT}_{512} \to \mathrm{ReLU} \qquad\qquad \text{size of } (512, 16, 16)$$
$$\to \mathrm{ConvT}_{256} \to \mathrm{BatchNorm} \to \mathrm{ReLU} \qquad \text{size of } (256, 32, 32)$$
$$\to \mathrm{ConvT}_{128} \to \mathrm{BatchNorm} \to \mathrm{ReLU} \qquad \text{size of } (128, 64, 64)$$
$$\to \mathrm{ConvT}_3 \qquad\qquad\qquad \text{size of } (3, 64, 64).$$

In all the $\mathrm{Conv}_k$ layers and all the $\mathrm{ConvT}_k$ layers except for the last, $5 \times 5$ convolutional filters with stride $(2, 2)$ are used. We use ReLU instead of leaky ReLU due to the performance consideration described in the previous subsection. To fit the size of the input images in our experiment, one extra convolutional layer is added for the encoder and the channel size is twice as large as that in Chen et al. (2016),

## K    EXAMPLES OF RECONSTRUCTED AND GENERATED IMAGES IN SECTION 5

We show examples of reconstructed images and images generated by sampling the learned approximated posterior from the proposed method and other works in Figures 4 and 5.

Figure 4: Reconstructed images and examples of images generated from the prior and the estimated posterior on MNIST. "GT" stands for ground truth.

## L    INTERPOLATION OF LATENT VARIABLES

This section aims to investigate the feasibility of the learned latent space of the methods mentioned in Section 5. If high quality images can be generated by interpolating the latent variables in a latent

| | Reconstructions | Random samples | | | |
|---|---|---|---|---|---|
| | | Prior | 2nd VAE | GMM-10 | GMM-100 |
| GT |  | | | | |
| VAE ($\sigma_x^2 = 1.0$) |  |  |  |  |  |
| WAE-MMD |  |  |  |  |  |
| AE |  | |  |  |  |
| RAE |  | |  |  |  |
| RAE-GP |  | |  |  |  |
| Iso-I w/ trainable $\Sigma_x$ |  |  |  |  |  |
| Diag-I w/ trainable $\Sigma_x$ |  |  |  |  |  |
| Iso-D w. trainable $\Sigma_x$ |  |  |  |  |  |
| Diag-D w/ trainable $\Sigma_x$ |  |  |  |  |  |
| Ours (Iso-I w/ AR-ELBO) |  |  |  |  |  |
| Ours (Diag-I w/ AR-ELBO) |  |  |  |  |  |
| Ours (Iso-D w/ AR-ELBO) |  |  |  |  |  |
| Ours (Diag-D w/ AR-ELBO) |  |  |  |  |  |

Figure 5: Reconstructed images and examples of images generated from the prior and the estimated posterior on CelebA.

space, the corresponding latent space is more likely to be feasible for other downstream tasks. Therefore, in this section, we evaluate the FID scores of images generated by latent variable interpolation for various models mentioned in section 5.

First, we choose 10,000 random pairs of images from both MNIST and CelebA datasets. The interpolation is done by applying spherical interpolation (Ghosh et al., 2020) in latent spaces and then generate the interpolated images with the decoders. In the end, we evaluate the FID of these interpolated images.

Furthermore, the experiment has been proceeded with two different setups of mixing ratios: (i) a fixed ratio of 0.5, i.e., the mid-point of two latent variables; and (ii) a uniformly distributed random ratio between $[0, 1]$ for each image pair. The result is shown in Table 4, where the proposed method achieved the best score on MNIST and is competitive on CelebA. This suggests that a generative model with proper smoothness achieved via the proposed method is also feasible for other applications such as interpolation and possibly applicable for other semantic controls.

| | MNIST | CelebA |
|---|---|---|
| VAE ($\sigma_x^2 = 1.0$) | | |
| WAE-MMD | | |
| AE | | |
| RAE | | |
| RAE-GP | | |
| Iso-I w/ trainable $\Sigma_x$ | | |
| Diag-I w/ trainable $\Sigma_x$ | | |
| Iso-D w/ trainable $\Sigma_x$ | | |
| Diag-D w/ trainable $\Sigma_x$ | | |
| Ours (Iso-I w/ AR-ELBO) | | |
| Ours (Diag-I w/ AR-ELBO) | | |
| Ours (Iso-D w/ AR-ELBO) | | |
| Ours (Diag-D w/ AR-ELBO) | | |

Figure 6: Examples of interpolated images.

