# OpenReview forum: "AR-ELBO: Preventing Posterior Collapse Induced by Oversmoothing in Gaussian VAE"
_ICLR.cc/2021/Conference — Reject_

### Official Review · AnonReviewer3 · 2020-10-28
**Idea is interesting but more explanations are required**

**Rating:** 6
**Confidence:** 4

**Review:**

This paper studies the Gaussian VAE and figures out that the decoder variance regularizes the VAE and affects the model smoothness, and an inappropriate estimation of this parameter would raise posterior collapse, which is supported by theoretical analysis and empirical demonstrations. Hence, this paper then proposes an ELBO with adaptive decoder variance to avoid oversmoothing the model. Overall, the idea is interesting and provides some new insights for our community. The major concerns regarding this paper are listed as below.

Since the variance parameter affects the performance of VAE, this paper proposes an adaptive training strategy using alternative updating of variance and the remaining parameters. In comparison to the strategy that directing treating the variance as a trainable parameter, though the following numerical experiments showcase that the proposed training strategy achieves higher FID scores, the reviewer is still unclear about the main difference between the two training strategies. The authors are suggested to make more explanations.

From the comparison results in Table 3, it seems that the parameterizations of variance have a large impact on the performance of the proposed VAE. As for VAE (sigma^2_x: optimizer) and Ours (Iso-I), their structures are the same with the only difference in training strategy. It is observed that VAE (sigma^2_x: optimizer) outperforms Ours (Iso-I) on the MNIST dataset.
Hence, the authors are suggested to test the conventional VAE using all the studied four parameterizations of variance in order to provide a comprehensive comparison.

---

> ### Author Response · Authors · 2020-11-13
> **Reply to reviewer 3**
>
> Thanks for your kind comments. Please find our reply below.
>
> *Since the variance parameter affects the performance of VAE, this paper proposes an adaptive training strategy using alternative updating of variance and the remaining parameters....*
>
> Regarding the difference between two strategies, since we are using MLE to adapt $\sigma_x^2$, it is optimized for the batch and it should be more stable than treating it as a training parameter.
>
> *From the comparison results in Table 3, it seems that the parameterizations of variance have a large impact on the performance of the proposed VAE....*
>
> In fact, upon receiving this comment, we performed an extra experiment, and it will be appended in the revised manuscript. This extra experiment tests the performances for the two strategies on all 4 parametrizations (Iso-I, Iso-D, Diag-I, Diag-D). The proposed strategy results into better FID and the strategy that treats variance parameters as usual training parameters do not work well in the Diag-D configuration on both MNIST and CelebA. This result supports the argument that the proposed method is more stable than the conventional one.

---

### Official Review · AnonReviewer4 · 2020-10-28
**A simple method, but with limited scope**

**Rating:** 4
**Confidence:** 3

**Review:**

This paper analyses the "posterior collapse" phenomenon observed in training latent variable models (in particular Variational Auto-Encoders), and propose a new training objective to remedy the problem. The theoretical analysis of the authors suggests that the posterior collapse is induced by an inappropriate choice of variance in the decoder distribution. The new objective they propose, the AR-ELBO, jointly optimises this variance along with the usual network parameters. The authors demonstrate that their objective yields relatively good results on image modelling, compared to other standard VAE methods.

I think the paper is generally well-written, with sufficient clarity around the major hypotheses and results. The motivation for the paper is also well established, with a sensible exposition of the problem of posterior collapse given through mutual information, which seems intuitive. Additionally, the theoretical analysis is straightforward to follow and explained relatively well, which is appreciated. The empirical results do also seem promising, with the metrics given demonstrating the proposed objective seems to be competitive with other widely used objectives for equivalent models.

I think the proposed objective is simple, which is a good thing. The variance of the output distribution is substituted by the current expected MSE of the reconstructions, which is its optimal value at any point in the optimisation. This results in the variance parameter becoming implicit. Table 1 shows the trade-off between MSE and KL nicely, and we can see the “sweet spot” of the choice of variance (to minimise the overall objective). There is an empirical comparison to the more obvious approach of just optimizing the variance parameter using gradient descent along with the neural network parameters. Something I would be interested in seeing discussed is why the AR-ELBO approach is a better method. I suspect that there may be something more concrete that can be said, in fact the substitution to eliminate the variance from the objective is reminiscent of the Rao-Blackwell theorem, which essentially states that it is best to integrate out variables in an estimator if you can.

I do however, have a number of problems with the paper in its current form. One major issue is the seemingly limited setting in which the analysis is provided. The analysis is only defined on a continuous data domain, with Gaussian output distributions. And in fact the core of the hypothesis and results centers on the premise that the width of this Gaussian is a crucial component of the optimisation. However, it is in practice quite rare to actually have continuous data. Almost all data we encounter is discrete, and even the limited datasets the authors assess their objective on is discrete - MNIST and CelebA - since image data is discrete. It is common (and surely best) practice with such image data to use a discrete output distribution p(x|z), for example a discretised mixture of logistic distributions, which has been demonstrated to be very successful. The authors only provide any results or analysis for the continuous data case, with Gaussian output distributions, and in fact do not even mention that the discrete case is the usual case, or that the data that they seek to model is actually discrete. This seems to be misleading, or at the very least very restrictive. Indeed, it seems that in the discrete output distribution case, the analysis the authors provide and the objective they propose does not hold, since it relies on an algebraic manipulation of the Gaussian pdf (or derivative of).

Another issue I have is that the experimental verification of the AR-ELBO deals only with the sample quality (of either reconstructions or true samples), but this is not the full picture of a VAEs performance. We also care about the structure of the latent space itself, for example to perform latent interpolations or downstream tasks with the latents. We also care about the approximate density function we have learned - to perform anomaly detection etc. These are not mentioned in the results at all, which I think is an omission.

Overall, I quite like the simple and well-presented nature of the paper, but the limited scope as raised above means I think the paper should be rejected in its current form. I would be willing to increase my score if the authors addressed some of the concerns I have raised.

---

> ### Author Response · Authors · 2020-11-13
> **Reply to reviewer 4**
>
> Thanks for your kind comments, the second one gives us a nice direction of an extra experiment. Please find our reply below.
>
> *One major issue is the seemingly limited setting in which the analysis is provided....*
>
> Regarding to the concern about the limitation of this work, it is true that digital data is discrete and in many cases they are categorical. However, at the same time, it is also a quantized result which is sampled from a continuous process in the real world. In many cases, assuming the generative process follows a continuous distribution will not pose significant negative effect (e.g. when the quantization step is small enough compared to the range of value). Also, since the discrete Binomial distribution approaches normal distribution (Gaussian) when N (the sample size) is large enough, it should be acceptable to use Gaussian directly as there are many useful mathematical tools for it.  We believe this is also one reason why so many papers (include some VAE papers) assumed Gaussian distribution when they have no other priors.
>
> On the other hand, it must be interesting to investigate what will happen if this approximation is not applicable. There is also Bernoulli VAE which is mentioned by another reviewer here. However, as a starting point, the Gaussian assumption is very helpful and has no loss of generality in many cases. By the way, the generality of the Gaussian setup in continuous domain is shown in “Diagnosing and enhancing VAE models”
>
> *Another issue I have is that the experimental verification of the AR-ELBO deals only with the sample quality (of either reconstructions or true samples), but this is not the full picture of a VAEs performance....*
>
> Regarding the second concern, we agree that it is better to have more inspection as suggested. Therefore, we will add additional experiment about the interpolation of latent vectors and use FID to access the quality of interpolation of different methods. The result will be added to the revised paper.

---

### Official Review · AnonReviewer1 · 2020-10-28
**Interesting analyses, limited novelty**

**Rating:** 6
**Confidence:** 4

**Review:**

AR-ELBO: Preventing posterior collapse induced by oversmoothing in Gaussian VAE

Summary:

The paper discusses the situation where the posterior approximation in the Gaussian variational autoencoder collapses during training. It is argued that a major contributing factor is a mismatched output variance parameter. Several analyses provide evidence for and intuition about this behavior. A simple procedure, where a maximum likelihood estimate of the noise is employed, is proposed. Experiments on some standard datasets provide an empirical evaluation of the method.

Positive:

1. The paper is in general well written and fairly easy to follow, at least for readers familiar with the Gaussian VAE.
2. The analyses are interesting and provide some good insights.
3. The proposed methods are presented in sufficient mathematical detail, including details for different variants of method.

Negative:

1. I encourage the authors to make the code available during the review process. Without access to the code, it is difficult to assess how well the results can be reproduced.
2. Since this is not the first work to address estimating the noise variance in a VAE setting, I would like to have seen a more direct comparison with competing methods. This could include results on how precisely the variance is estimated and how the procedure influences the convergence of other parameters.
3. The novelty of the proposed method is fairly limited - I would expect that learning the noise is common practice in applied work.

Recommendation:

Weak accept

Further comments:

"Contrary to popular belief..." could you provide a reference?

"However, in most implementations ... constant of 1.0." is this really true? I would assume most people would either choose the noise variance based on prior knowledge or fit it (with maximum likelihood or a variational approximation) long with the other model parameters. If it really is common practice to not set the noise in accordance with the data, then the procedure presented in this work is definitely a much better default.

The paper uses the term "adaptive" - I would prefer to phrase this as maximum likelihood.

---

> ### Author Response · Authors · 2020-11-13
> **Reply to reviewer 1**
>
> Thanks for your kind comments. Please find our reply below.
>
> *I encourage the authors to make the code available during the review process....*
>
> We agree the aspect regarding the reproducibility. We are proceeding to get authorization from our organization to make the source code at least available to all reviewers within the discussion period.
>
> *Since this is not the first work to address estimating the noise variance in a VAE setting, I would like to have seen a more direct comparison with competing methods.....*
>
> Regarding the precision of variance estimation. The best value of $\sigma_x^2$ depends on the data space formed by all generated samples. Since the model parameters keep changing, the best value also changes with every parameter update. Since we have no access to the true data distribution of the generated samples, we can only “estimate” $\sigma_x^2$ through the samples, which leads us to the maximum likelihood estimation. However, we’re also working on an extra experiment which treats $\Sigma_x$ as a usual learnable parameter (the conventional method) in different parametrizations (Iso-D, Diag-I, Diag-D). This extra experiment should show the difference between the proposed method and the conventional one and serve as a direct comparison to existing methods. The experiment will be appended to the revised paper.
>
> *The novelty of the proposed method is fairly limited - I would expect that learning the noise is common practice in applied work.*
>
> Estimating the strength of noise is indeed common in some conventional applications (e.g. L-curve, Generalized cross validation for the linear case). However, here we also provide a comprehensive insight about the relation between posterior collapse and the smoothness of a learned non-linear function. Besides, we show its link to the upper-bound of mutual information. Moreover, corresponding objective functions for various parametrizations other than the standard one are derived. Starting from the Diag-D parametrization, we also would like to investigate the methodology to deal with the most general parametrization where the variances are model by a generic covariance matrix. In an extra experiment which will be appended to the revised manuscript, AR-ELBO shows better stability over the conventional approach (the approach that treating $\sigma_x$ as training parameter directly) in the Diag-D case.
>
> *“Contrary to the popular belief...” could you provide a reference?*
>
> The reference is Lucas et al. (2019) (please also see Figure 1 of this work), we will fix the reference in the revised manuscript.
>
> *"However, in most implementations ... constant of 1.0." is this really true?....*
>
> We will use a more precise description, such as “a constant which is data independent” in the revised version. Still, we can enumerate several examples:
> 1.github.com/PacktPublishing/Advanced-Deep-Learning-with-Keras/blob/master/chapter8-vae/
> 2.github.com/ritheshkumar95/pytorch-vqvae/
>
> This implementation makes sigma a fixed hyperparameter with default value=0.001:
> 3.github.com/psanch21/VAE-GMVAE/tree/master/Alg_VAE
>
> These are all great implementations and people refer to these implementations from time to time. However, they assume $\sigma_x^2$ as a constant that is independent of the input data samples.
>
> *The paper uses the term "adaptive" - I would prefer to phrase this as maximum likelihood.*
>
> Although we use maximum likelihood estimation to determine $\sigma_x^2$ in the ELBO, it is in fact determining $\sigma_x^2$ adaptively per update. Since $\sigma_x^2$ affects the strength of regularization, we therefore use “adaptively regularized” as the name. We also wonder that if we use a name such as MLE-ELBO, perhaps it will make some people misunderstood that we’re combining MLE and ELBO to form a new objective (the proposed AR-ELBO is still an ELBO-like loss).

---

> > ### Comment · AnonReviewer1 · 2020-11-19
> > **Thank you for your reply**
> >
> > I appreciate that you have made the code available, and include an experiment with the noise treated as a usual parameter.

---

### Official Review · AnonReviewer2 · 2020-10-29
**Review of paper #2673**

**Rating:** 7
**Confidence:** 4

**Review:**

The paper considers Gaussian VAEs and their tendency to suffer from posterior collapse. In particular, the authors analyse the impact of the usually fixed covariance $\sigma_x$ of the decoder Gaussian on the learned encoder variance. They show that the former can be seen as a regulariser for the latter and therefore impacts the "smoothness" of the encoder. The authors hypothesize that a large value of $\sigma_x$ causes posterior collapse as a consequence.

As a remedy, the authors propose to consider the decoder covariance as a learnable parameter. To achieve this, they propose to optimize the ELBO objective by a block-coordinate descent approach where the parameters of the decoder covariance are considered as a separate block. The possibly remaining prior-posterior mismatch in the latent distributions is mitigated by an second stage VAE as e.g. in Dai and Wipf, 2019. Experiments compare the new approach (on MNIST and CelebA) with existing approaches in terms of MSE on the training data and Frechet inception distance for the generator and show that it is least on par or outperforming them.

The paper is well written and technically correct. All necessary concepts are concisely introduced. Its novelty is in my view the following:
(1) the new, stronger definition of posterior collapse
(2) the analysis of the regularising impact of the decoder variance
(3) the proposed method for leaning the parameters of decoder variance

On the downside, I am not convinced that posterior collapse is caused solely by over-smoothness of the encoder. The reasons are the following.
(1) The analysis in Dai and Wipf, 2019, Theorem 2 presumes that the encoder covariance is not required to be diagonal.
(2) Posterior collapse is also observed in Bernoulli VAEs, where the latent variables are binary valued vectors. In these models, usually all(!) parameters of the encoder/decoder are learned by optimising the ELBO objective.

*Further comments*
Please describe the relation of your objective (9) to $\beta$-VAEs.

The authors have tried to answer all issues and questions raised in the reviews. At least, I can say so for the questions and issues raised by me. I therefore tend to keep my positive opinion on this paper.

---

> ### Author Response · Authors · 2020-11-13
> **Reply to reviewer 2**
>
> Thanks for your kind comments. Please find our reply below.
>
> *On the downside, I am not convinced that posterior collapse is caused solely by over-smoothness of the encoder.*
>
> We do also aware that posterior collapse has other causes, the revised manuscript will clarify about this and emphasize that we are focusing on the posterior collapse caused by over-smoothness.  In fact, we feel that over-smoothing is a problem that should receive more attention, as it could happen in other variants of generative models as well.
>
> *The analysis in Dai and Wipf, 2019, Theorem 2 presumes that the encoder covariance is not required to be diagonal.*
>
> It is true that the encoder covariance can be just an arbitrary covariance matrix. This is the most general case and worth to derive corresponding theories/frameworks. However, due to its complexity, as far as we know, there’s no existing VAE-like models have treated this setup yet.
>
> *Posterior collapse is also observed in Bernoulli VAEs, where the latent variables are binary valued vectors....*
>
> Regarding the Bernoulli VAE, although we haven’t investigated about this setup yet, we will put this in the scope of our next step. It must be interesting to see how the latent space is regularized in the Bernoulli VAE and how it gets collapse.
>
> *Please describe the relation of your objective (9) to beta-VAEs*
>
> Although both (9) and beta-VAE contain the balancing parameter, they are conceptually different. We proposed to dynamically adapt $\sigma_x^2$ with MLE in a way that still compatible with the variational inference framework. On the other hand, beta-VAE fixes the balancing parameter with "$\beta>1$". Both works suggested that such parameter can affect the upper bound of mutual information between the latent space and the data space. We will make this more clearly in the revised paper.

---

### Author Response · Authors · 2020-11-18
**Response to all reviewers: Manuscript update**

We would like to thank again to all reviewers for their constructive comments, which helped us to improve our work. We have updated the manuscript. The details of the revision are as follows:


Two Extra Experiments
- Treat $\Sigma_x$ as usual trainable parameters on 4 parametrizations (Iso-I, Iso-D, Diag-I and Diag-D)
  - Table 3 in Section 5; Figures 4 & 5 in Appendix K.

- FID result of the latent space interpolation
  - Table 4 in Section 5 and Appendix L

Manuscript
- Introduction
  - Clarified that posterior collapse has other causes than oversmoothness
  - Added a citation for clarity ("contrary to the popular belief")
  - In the "contribution" part, added extra description to state that one should estimate $\sigma_x^2$ from data, instead of tuning it as a hyperparameter.

- Conclusion
  - Clarified the contributions and reflected the result of the extra experiments.

- Related Works (Appendix I)
  - Extra paragraph about the relation between the proposed method and beta-VAE has been added.

Thanks again for all the reviewers and chairs. We’re looking forward to your further comments.

---

### Decision · Program_Chairs · 2021-01-07
**Final Decision**

**Decision:**

Reject

**Comment:**

This paper is about learning the output noise variance of a VAE and its effect on the generated image quality as measured by FID. The paper argues that the output variance parameter plays an important role and proposes a simple procedure, where a maximum likelihood estimate of the noise variance is estimated. Experiments on some standard datasets are provided.
Overall, the paper is well written and has been perceived positively by the reviewers. However, the effect of observation variance has been in detail analysed by earlier work, in particular Dai and Wipf 2019. The novelty of the current paper is somewhat limited in scope. The paper is somewhat borderline in these respect; a much stronger experimental section would have been helpful.

One key contribution of the work is empirical comparison of alternative parametrizations of the output noise. Overall, the paper would be stronger if this aspect is analysed more in detail, possibly with careful comparisons with competing methods. Inclusion of controlled experiments (e.g. by adding extra noise to data) to show how precise the noise variance estimation and how the procedure influences the convergence of other parameters would have made the paper much more impactful.